# Beyond Policy Training: Recursive Solution Search from Unannotated Videos

Lipeng Wan [* 1]   Jianhui Gu [* 1]   Junjie Ma [1]   Anbang Wang [1]   Xuguang Lan [1]

## Abstract

Many real-world tasks are recorded as large collections of unannotated task executions, such as videos, which contain rich information about task progress but lack the supervision assumed by standard reinforcement learning (RL) pipelines. In many practical settings, the goal is not to train a reusable policy but simply to recover one feasible solution, making policy-centered learning unnecessarily costly. We propose Policy-Free Recursive Search (PFR-Search), a framework that directly recovers solutions from unannotated task executions without policy-grounded supervision or policy training. PFR-Search organizes videos into high-level task flows and performs recursive backward-forward search to recover solutions without policy modeling. To evaluate the efficiency of policy-free search in exploiting unannotated data, we use RL as an evaluation interface, incorporating task-flow-aligned intrinsic rewards and compare against video-driven RL methods. Experiments on long-horizon Minecraft tasks show that PFR-Search recovers feasible solutions from unannotated videos with minimal exploration.

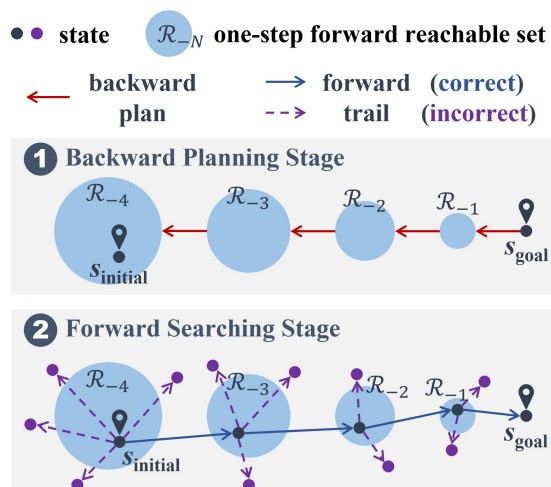

*Figure 1.* Conceptual prototype for solution recovery from unannotated task executions. The prototype reasons recursively through a backward–forward process. First, backward planning starts from the goal and infers which intermediate stages must precede it, forming a coarse high-level roadmap. Second, forward search begins at the start state and follows this roadmap to guide environment interaction toward the goal. Correct trajectories remain aligned with the inferred roadmap, while incorrect trials deviate when their visited states no longer follow the inferred stage sequence.

## 1. Introduction

Through training executable policies from interaction and annotated data, reinforcement learning (RL) has achieved impressive success across many decision-making tasks (Kaelbling et al., 1996). However, policy training remains constrained by the demand for large quantities of high-quality data (Ma et al., 2025a; Ball et al., 2023; Ma et al., 2024). Such data must be obtained either through extensive environment interaction, which is noisy and inefficient, or through human annotation, which is costly and difficult to scale. In many realistic scenarios, the goal is not to obtain a reusable policy but simply to discover one feasible solution, which makes policy learning unnecessarily expensive. In contrast, large repositories of unannotated task executions (e.g., videos) are readily available, containing rich information about task progress and structure but lacking the policy-grounded supervision assumed by standard RL pipelines (Minderer et al., 2019; Epstein et al., 2021; McCarthy et al., 2025). This mismatch highlights the absence of a mechanism for directly recovering solutions from unannotated task executions without policy-grounded supervision and policy learning.

Recent progress has shown that unannotated videos can assist decision making through representation pretraining, reward shaping, or task structure discovery (Baker et al., 2022; Ma et al., 2023; Luo et al., 2020). However, existing approaches typically rely on annotated or interaction-generated data to train models that extract supervision from unannotated videos (Lifshitz et al., 2023; Miech et al.,

*Equal contribution [1]Xi'an Jiaotong University, Xi'an, China. Correspondence to: Xuguang Lan <xglan@xjtu.edu.cn>.

*Proceedings of the 43rd International Conference on Machine Learning*, Seoul, South Korea. PMLR 306, 2026. Copyright 2026 by the author(s).

2019; Dvornik et al., 2023). Compared to standard reinforcement learning, these methods reduce the amount of required annotation, but still require substantial annotated data to learn environmental dynamics, reward functions, or procedural predictors that are used to supervise downstream policy learning. Consequently, these works primarily use unannotated videos as auxiliary supervision alongside annotated data to assist policy learning, rather than to directly recover solutions from unannotated task executions (Ma et al., 2025a; Narvekar et al., 2020).

In response to this gap, we explore a different perspective: **how to recover solutions directly from unannotated task executions without policy learning?** Unannotated executions contain progression structure that reflects how tasks advance over time. We therefore consider a simple conceptual prototype that operates on this structure, as illustrated in Fig. 1. The prototype assumes access to a *task flow* extracted from unannotated videos, which captures high-level stages of task progress and their connectivity across executions. A backward planner reasons over this task flow, starting from the goal and inferring which stages must precede it, with the objective of identifying a high-level path from the goal back to the start. This backward reasoning produces a coarse roadmap of what must be achieved before what. Given this roadmap, forward search is then performed at the step level, following the inferred flow path to guide environment interaction and progressively construct a concrete solution trajectory. By anchoring step-level exploration to this high-level task flow, the prototype decomposes long-horizon solution recovery into a sequence of tractable local transitions.

This prototype naturally matches unannotated videos, which provide state-only execution sequences that reflect task progress without action or reward supervision. We instantiate this prototype as Policy-Free Recursive Search (PFR-Search) by extracting task flows from videos and performing planning directly over the resulting flows. PFR-Search operates without action or reward supervision and does not rely on policy learning. We use a small warm-start set of 50 clips with coarse stage labels only to initialize task-flow learning. For broader empirical evaluation of the efficiency of PFR-Search in exploiting unannotated data, we further use reinforcement learning as an evaluation interface. Specifically, we use the task-flow abstraction to define task-flow-aligned intrinsic rewards for reinforcement learning, which we refer to as PFR-guided RL. Under this interface, we compare PFR-Search with video-driven reinforcement learning methods in terms of how effectively unannotated videos are translated into environment interaction and task progress.

The key contributions of this work are as follows. **1**. We show that solutions to long-horizon tasks can be recovered directly from unannotated task executions without policy training. **2**. We develop PFR-Search, a policy-free recursive search framework that performs efficient solution recovery over video-derived task flows. **3**. We demonstrate that PFR-Search can recover feasible solutions with minimal exploration, and that its integration with reinforcement learning can further improve exploration efficiency.

## 2. Background

### 2.1. Policy-Grounded Decision Making

Sequential decision making is predominantly formulated as learning a control policy that maps states or histories to actions. This paradigm encompasses Reinforcement Learning (RL), imitation learning, and sequence-model-based approaches such as decision transformers. Despite algorithmic differences, these methods share a common objective: acquiring a reusable policy or policy-equivalent action model that can be executed in an environment to generate behavior.

RL formalizes this objective through interaction-driven policy optimization in a Markov Decision Process (MDP) $(\mathcal{S}, \mathcal{A}, p, r, \gamma)$ (Sutton & Barto, 2018), where the agent learns a policy $\pi(a \mid s)$ to maximize expected cumulative reward. Imitation learning and sequence-based models replace online interaction with fixed datasets or conditional generative objectives, but remain grounded in state–action trajectories and aim to train an executable policy.

A defining characteristic of this paradigm is its reliance on policy-grounded supervision. Whether through rewards, action labels, or return-conditioned trajectories, learning is anchored to signals that explicitly supervise actions, requiring annotated demonstrations, preference signals, or environment interaction to construct training objectives. As a result, this formulation presumes that solving a task entails learning a reusable policy, and offers no mechanism for directly exploiting large collections of unannotated task executions to recover solutions.

### 2.2. Task Knowledge Extraction from Unannotated Videos

Unannotated videos provide abundant task execution demonstrations as state-only trajectories that can potentially support decision making. However, due to the absence of action labels or reward supervision, they cannot be directly used for policy training.

A growing body of work investigates how unannotated videos can be exploited for decision making, including latent dynamics pretraining (Ma et al., 2023; Lifshitz et al., 2023), representation and reward learning (Escontrela et al., 2023; Liu et al., 2026), and task structure dis-

covery (Dvornik et al., 2023; Zhao et al., 2026). Despite their differences, these approaches ultimately aim to train executable policies, and rely on substantial annotated or interaction-generated data to train models that extract supervision from unannotated videos. As a result, unannotated videos in prior work primarily serve to complement supervised or interaction-driven learning, rather than to directly recover solutions from unannotated task executions. A more comprehensive discussion of related work is provided in Appendix A.

This distinction also separates PFR-Search from visual planning or continuous latent-space planning. Rather than predicting fine-grained future observations or continuous trajectories, PFR-Search deliberately plans over discrete task-flow categories, using them as a structural horizon decomposition that reduces long-horizon solution recovery into a finite sequence of local reachability problems. In this work, we shift the role of unannotated videos from auxiliary supervision to the central source of task knowledge for problem solving. Rather than using videos to derive signals for policy training, we investigate how to recover solutions directly over information extracted from unannotated task executions.

## 3. Methodology

We study the problem of recovering concrete solution trajectories directly from unannotated task executions without learning a control policy. To recover solutions without policy learning, we require a task-level space that supports reasoning about task progress, which motivates backward-forward reasoning over induced task flows. We instantiate this idea as Policy-Free Recursive Search (PFR-Search). Accordingly, PFR-Search first constructs a discrete task-flow space that captures coherent task progressions and their structural relations across demonstrations. This task-flow space defines the domain in which backward planning is performed.

The overall method therefore consists of two conceptual components. Section 3.1 describes how the task flow is abstracted from unannotated task executions. Section 3.2 presents the policy-free recursive backward-forward search algorithm. Section 3.3 further shows how the task-flow structure can be integrated with standard reinforcement learning.

### 3.1. Task Flow Abstraction from Unannotated Videos

We view unannotated videos as recordings of task progression, where each execution reflects a directed and largely irreversible evolution of task state, and different demonstrations share an underlying progression structure. To make such structure recoverable from unannotated data, the

learned abstraction must exhibit (i) coherent temporal progression within each execution and (ii) consistent structural alignment across executions. These two requirements motivate two complementary constraints: in-flow consistency and cross-flow consistency.

To operationalize this, we organize video observations into discrete task-flow categories, as illustrated in Fig. 2. Given a dataset of unannotated videos, each frame is encoded into a latent vector by a VQVAE, and clustered using DB-SCAN to produce discrete task-flow categories. In parallel, we train an autoregressive backward planner (BP), a category-transition model that predicts predecessor relations between task-flow categories. At this stage, BP is used to impose cross-demonstration predecessor consistency during task-flow learning; after the task-flow abstraction is learned, the same BP is used for backward planning in Section 3.2. We use a small warm-start set with coarse stage labels only to initialize the VQVAE, clustering, and backward planner. Together, they shape the latent space so that categorical task flows form stable abstractions that directly support recursive backward-forward solution search.

**In-flow consistency.** A single video provides an ordered execution of a task, where states evolve along a largely irreversible progression. To make such progression structure recoverable from unannotated data, we require that the latent trajectory exhibits a coherent directional trend, reflecting monotonic task advance within each execution.

Given the latent sequence $\{z_1, \ldots, z_T\}$ of a video, we instantiate in-flow consistency as a directional alignment constraint. We define a global task direction

$$\text{flow}_{\text{vec}} = z_T - z_1.$$

Each local transition $(z_{j+1} - z_j)$ is encouraged to align with this global direction, so that the latent trajectory consistently progresses rather than oscillating or reversing.

Directional alignment is enforced via:

$$\mathcal{L}_{\text{in-flow}} = \frac{1}{T-1} \sum_{j=1}^{T-1} \big(1 - \cos(z_{j+1} - z_j,\ \text{flow}_{\text{vec}})\big). \quad (1)$$

**Cross-flow consistency.** Different demonstrations of the same task exhibit a shared progression structure. To align their task flows, we use an autoregressive backward planner (BP) that models transitions between flow categories. Let $P(c_k \mid z_{t,i})$ be the probability of assigning latent $z_{t,i}$ to category $c_k$, and let $t_{m \to k}$ denote the BP-estimated probability that category $k$ is the predecessor of $m$ in a backward transition. Cross-flow consistency penalizes deviations from

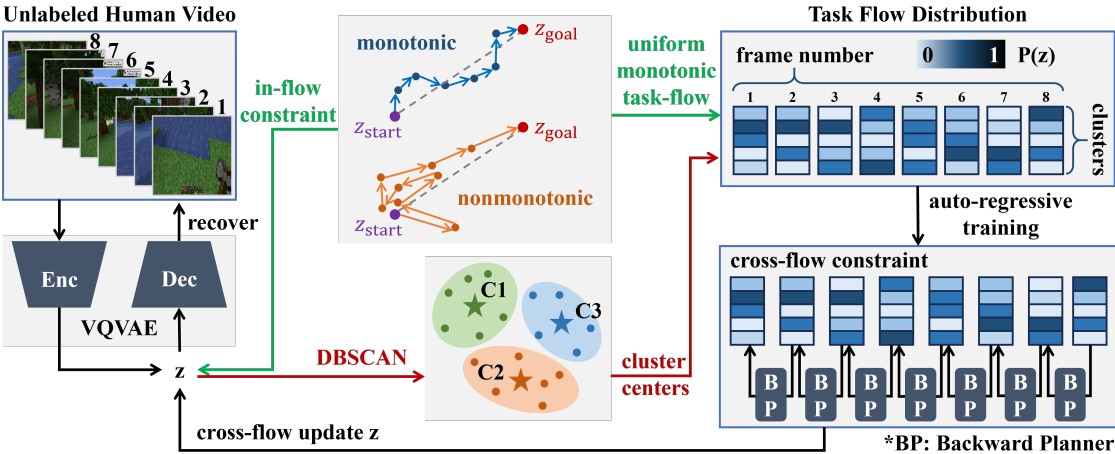

*Figure 2.* Constructing task-flow abstractions from unannotated task executions. Unannotated video frames are encoded by a VQVAE, whose latent space is refined by reconstruction loss, in-flow consistency, and cross-flow consistency. DBSCAN clustering produces discrete flow categories $(C_1, C_2, C_3)$, and an autoregressive backward planner (BP) models category transitions across demonstrations. These two consistencies induce task-flow abstractions that capture coherent progression within each execution and consistent structure across demonstrations.

this autoregressive structure:

$$\mathcal{L}_{\text{cross-flow}} =$$
$$\sum_{i=1}^{N} \sum_{t=2}^{T_i} \sum_{k=1}^{K} P(c_k \mid z_{t-1,i}) \log \frac{P(c_k \mid z_{t-1,i})}{\sum_{m=1}^{K} P(c_m \mid z_{t,i}) \, t_{m \to k}}. \quad (2)$$

This objective encourages multiple demonstrations to fit into a consistent and transferable task-flow structure. In-flow consistency alone could encourage representations that correlate with within-video temporal position. Cross-flow consistency mitigates this risk by requiring predecessor relations to be shared across demonstrations with different speeds, lengths, and visual details, thereby favoring transferable task-progress structure rather than clip-specific temporal shortcuts.

**Total loss.** The overall objective integrates reconstruction fidelity and both consistency constraints:

$$\mathcal{L}_{\text{total}} = \mathcal{L}_{\text{rec}} + \alpha \, \mathcal{L}_{\text{in-flow}} + \beta \, \mathcal{L}_{\text{cross-flow}}, \quad (3)$$

where $\alpha, \beta > 0$ balance the contributions.

### 3.2. Policy-Free Recursive Solution Search

Given a learned task-flow abstraction, PFR-Search recovers a concrete solution trajectory by performing recursive search over task stages, without learning or executing any control policy. Solution recovery is decomposed into two coupled operations: (1) backward planning in task-flow space to infer an ordered sequence of intermediate stages that connect the goal to the start, and (2) forward search in the environment to realize these stages and construct a concrete solution trajectory.

**Backward planning.** Backward planning operates in task-flow space and is responsible for constructing a high-level structural plan. As illustrated in Fig. 3, the initial image $s_{init}$ and goal image $s_{goal}$ are first encoded into latent vectors $z_{init}$ and $z_{goal}$ using the VQVAE encoder. These latents are mapped to their corresponding flow categories $c_{init}$ and $c_{goal}$ via cluster assignment rule. Starting from $c_{\text{goal}}$, the backward planner (BP) iteratively predicts predecessor categories by outputting a distribution over all possible predecessors at each step; we ignore the self-category and sample from the remaining distribution. By iteratively applying one-step predecessor prediction from $c_{\text{goal}}$ until $c_{\text{init}}$ is reached, we obtain a high-level task-flow plan $TF = \{c_0, c_1, \ldots, c_N\}, \quad c_0 = c_{init}, \; c_N = c_{goal}$.

**Forward searching.** Forward search is responsible for realizing the planned task-flow sequence and constructing a concrete solution trajectory. Fig. 4 shows how the agent executes the planned task flow. Since PFR-Search does not learn a policy, forward search uses random actions, with guidance provided only by the high-level task-flow plan. To reach each successive category transition $(c_i \to c_{i+1})$, the agent performs a *one-flow search*: it interacts with the environment using random actions and continuously encodes its current state to check whether its flow category matches the target $c_{i+1}$. All visited states and actions before reaching $c_{i+1}$ form a *flow segment*

$$\tau_i = \{(s_{i,0}, a_{i,0}), (s_{i,1}, a_{i,1}), \ldots\}.$$

The first state that matches $c_{i+1}$ is recorded as the anchor state $s_{i+1,0}$. During a one-flow search, we monitor the agent's current flow category at every step. If the agent

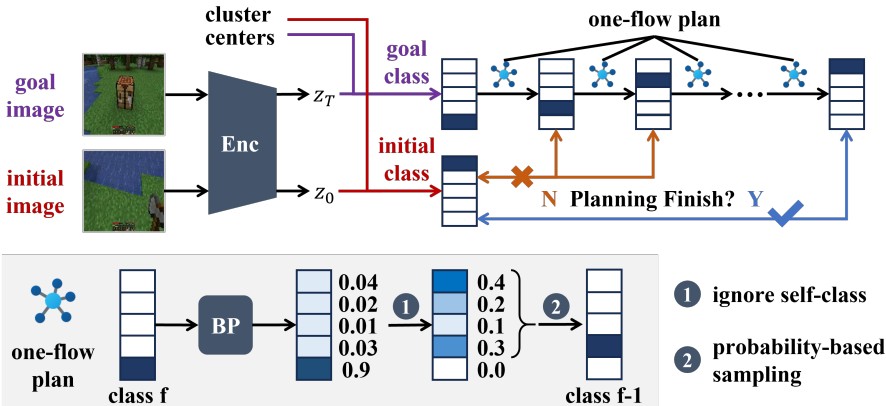

*Figure 3.* Backward planning in PFR-Search. The initial and goal images are encoded into latent vectors $(z_{\text{init}}, z_{\text{goal}})$ and mapped to their corresponding flow categories $(c_{\text{init}}, c_{\text{goal}})$. The backward planner then generates a task-flow plan by iteratively predicting predecessor categories: at each step, it outputs a distribution over possible predecessor flows and samples from it while excluding the self-category. This iterative process continues until the initial flow category is reached, yielding a complete high-level task-flow plan from goal to start.

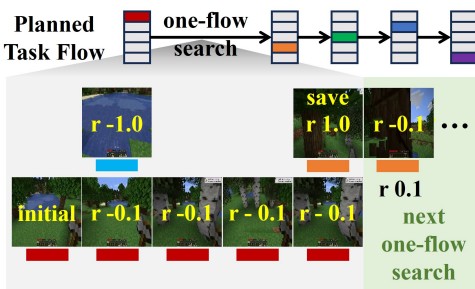

*Figure 4.* Forward search in PFR-Search. Given the task-flow plan produced by backward planning, the agent performs iterative one-flow searches from $c_i$ to $c_{i+1}$. Each search begins at the current anchor state and terminates when either the next flow category is reached or the step limit is exceeded. The first encountered state belonging to the next planned category is recorded as the new anchor state. Flow-segment replay enables progress preservation without environment rollback. The figure also shows the reward structure used when extending the method to PFR-guided RL.

transitions to a category that is neither the current $c_i$ nor the next $c_{i+1}$, we define this as a deviation from the planned flow. In this case, the trial is terminated and the agent returns to the previous anchor state $s_{i,0}$ by replaying the already discovered flow segments $\{\tau_0, \ldots, \tau_{i-1}\}$. The next one-flow search then begins from $s_{i,0}$ and proceeds until $s_{goal}$ is reached. Replay is implemented by re-executing the previously discovered action sequences from the environment initial state, and does not assume environment rollback or arbitrary state setting. In our implementation, replay is deterministic: we fix the environment seed and re-execute the saved action sequence from the same initial condition, so the previously discovered category sequence is reproduced in our experiments.

Forward searching decomposes an otherwise intractable

long-horizon task into a sequence of tractable local transitions. This decomposition is induced by the backward-planned task flow, which provides a semantic progression standard, and is operationalized through anchor-based replay, which retains successful partial progress across failed attempts. By reusing previously discovered flow-aligned segments, the agent accumulates progress monotonically along the planned flow without environment resets, thereby avoiding redundant exploration and enabling efficient policy-free task completion. Removing either task-flow guidance or anchor-based replay eliminates this decomposition and collapses the procedure to unguided random exploration.

### 3.3. Task-Flow Guided Reinforcement Learning

While PFR-Search focuses on policy-free solution recovery, the learned task-flow abstraction also provides a structural prior for standard reinforcement learning. We therefore introduce PFR-guided RL, which instantiates the extracted task-flow abstraction within a standard policy optimization pipeline through flow-aligned intrinsic rewards.

Given a task flow $TF = \{c_0, \ldots, c_N\}$, PFR-guided RL defines a shaped reward that reflects progress along the flow. When the agent first reaches a state whose category matches the next planned flow $c_{i+1}$, it receives a positive reward $r_{\text{anchor}}$. If the agent enters a category that deviates from the current target $c_{i+1}$, it receives a negative reward $r_{\text{off}}$. At every step, a small step penalty $r_{\text{step}} < 0$ is applied to discourage unproductive wandering.

For each task, we first learn a task-flow abstraction from unannotated videos under the proposed in-flow and cross-flow consistency constraints, and extract a set of task flows to define task-flow-aligned intrinsic rewards. PFR-guided

RL then explores the environment under the oracle-flow shaped reward, without backward planning, anchor states, or replay. It serves as an evaluation interface for testing whether the learned task-flow abstraction provides actionable exploration signals, rather than as the policy-free solution-recovery procedure itself. The policy is trained to maximize this task-flow-aligned intrinsic reward using standard reinforcement learning. In this way, PFR-guided RL evaluates whether the task-flow abstraction extracted from unannotated videos provides a stable and actionable exploration signal for long-horizon tasks.

## 4. Experiments

### 4.1. Experimental Setup

Our experiments are organized around three questions: (1) whether task-flow abstractions learned primarily from unannotated videos are sufficient to support policy-free solution recovery in long-horizon tasks; (2) whether the proposed in-flow and cross-flow consistency objectives effectively promote such solution recovery; and (3) whether the task-flow abstraction improves exploration when used to define intrinsic rewards for reinforcement learning.

We first report the success rate of backward planning and forward searching, together with comparisons to unguided random exploration and flow reachability. We then ablate the in-flow and cross-flow consistency objectives to assess their roles in constructing task-flow abstractions. Finally, we evaluate PFR-guided RL to examine how task-flow extracted from unannotated videos can guide exploration in policy learning. We use a small warm-start set of 50 clips with coarse stage labels only to initialize task-flow learning for both PFR-Search and PFR-guided RL.

**Benchmark.** We require a benchmark with abundant unannotated videos, multi-stage long-horizon tasks, and sparse progress signals to test whether video-derived task-flow structure is actionable. Minecraft satisfies these criteria, offering diverse multi-step tasks and large-scale egocentric videos. Task-flow learning is performed on 1200 unannotated clips from the OpenAI Contractor Dataset (Baker et al., 2022), covering five tasks (Harvest water, Harvest log, Harvest sand, Mine cobblestone, Mine iron ore).

**PFR-guided RL and baselines.** In the reinforcement learning setting, we build PFR-guided RL on PPO (Schulman et al., 2017) using task-flow-aligned intrinsic rewards. We compare PFR-guided RL with representative video-pretrained policies (VPT (Baker et al., 2022), STEVE-1 (Lifshitz et al., 2023), PTGM (Yuan et al., 2024)) and reward-shaped RL methods (Director (MineCLIP), Director (PPO) (Fan et al., 2022)). All methods are evaluated

under the same downstream interaction budget of $10^6$ environment steps per task, using their standard reward formulations. This comparison does not equalize total pretraining data or compute: several baselines rely on substantially larger or more specialized pretraining resources, such as Minecraft-scale video pretraining, instruction-related supervision, or pretrained MineCLIP reward models. We do not compare with model-based RL methods because PFR neither learns nor uses a dynamics model. PFR-Search operates purely over the video-derived task-flow abstraction rather than predicting environment transitions. Implementation details and fairness discussions are in Appendix C and Appendix B, respectively.

### 4.2. Evaluation of Backward Planning and Forward Searching

We evaluate backward planning and forward searching to assess whether task-flow abstractions learned from unannotated videos are sufficient to support policy-free solution recovery in long-horizon tasks.

*Table 1.* Backward planning performance across five long-horizon Minecraft tasks.

| Task | Success Rate (%) | Plan Steps (Success) |
|---|---|---|
| Harvest water | 36.0 | 4.15 ± 2.03 |
| Harvest log | 62.0 | 5.33 ± 1.77 |
| Cobblestone | 78.0 | 5.05 ± 1.88 |
| Iron ore | 47.0 | 5.03 ± 1.84 |
| Harvest sand | 37.0 | 6.53 ± 1.75 |
| Average | 52.0 | 5.26 ± 1.85 |

**Backward planning success rate.** We measure planning success as the ability to recover a valid path from the goal category back to the initial category within 10 one-flow steps. Tab. 1 reports success rates and the length of successful plans. The planner is most reliable on tasks with stable visual structure and predictable stage transitions (e.g., Mine cobblestone, 78.0%). Tasks such as Harvest sand or Harvest water exhibit larger appearance variation and ambiguous intermediate states, leading to lower success rates.

*Table 2.* Backward planning success rates with 50-sample warm-up vs. no warm-up

| Task | 50-sample warm-up | no warm-up |
|---|---|---|
| Harvest water | 36.0 | 24.0 |
| Harvest log | 62.0 | 32.0 |
| Cobblestone | 78.0 | 35.0 |
| Iron ore | 47.0 | 21.0 |
| Harvest sand | 37.0 | 12.0 |
| Average | 52.0 | 24.8 |

Across all tasks, successful plans contain around 5 semantic steps, each corresponding to a high-level flow transition that typically spans dozens or hundreds of environ-

*Table 3.* Forward searching performance across five long-horizon Minecraft tasks.

| Task | Oracle Flow | | | End-to-End | | | Random |
|------|-------------|---|---|------------|---|---|--------|
| | **Success Rate (%)** | **Trajectory Len.** | **Replay Times** | **Success Rate (%)** | **Trajectory Len.** | **Replay Times** | **Success Rate (%)** |
| Harvest water | 35.0 | 479 ± 122 | 9.84 | 13.0 | 513 ± 136 | 8.31 | 1.0 |
| Harvest log | 13.0 | 356 ± 75 | 6.20 | 8.0 | 391 ± 89 | 7.17 | 1.0 |
| Cobblestone | 57.0 | 502 ± 117 | 11.53 | 44.0 | 480 ± 114 | 12.10 | 5.0 |
| Iron ore | 11.0 | 568 ± 94 | 4.75 | 6.0 | 612 ± 133 | 6.32 | 0.0 |
| Harvest sand | 23.0 | 376 ± 131 | 8.35 | 9.0 | 388 ± 127 | 11.26 | 2.0 |
| Average | 27.8 | 456.2 | 8.13 | 16.0 | 477 | 9.03 | 1.8 |

ment steps. Tab. 2 further evaluates the effect of the 50-sample warm-start set used for task-flow initialization. Removing the warm-start set decreases the average backward-planning success rate from 52.0% to 24.8%, indicating that coarse warm-start labels substantially stabilize task-flow learning. At the same time, non-zero performance without warm-start clips suggests that useful task-progress structure can still be extracted from unannotated videos alone. We further analyze sensitivity to the amount of unannotated video data in Appendix D.2. Reducing the video set from 1200 to 600 and 300 clips decreases the average backward-planning success rate from 52.0% to 43.4% and 24.0%, respectively, suggesting that PFR-Search benefits from sufficient task-progress coverage while degrading predictably as coverage decreases.

**Forward searching success rate.** We evaluate forward search by comparing three settings: unguided random exploration, end-to-end PFR-Search using learned task flows, and oracle forward search guided by task-flow sequences extracted from successful human demonstrations. Results are summarized in Tab. 3.

Unguided random exploration achieves only 1.8% average success across tasks. In contrast, end-to-end PFR-Search recovers solutions in 16.0% of trials on average, showing that task-flow abstractions learned from unannotated videos provide actionable structure for policy-free solution recovery. When oracle task flows are provided, success rates further increase to 27.8%, indicating that the remaining performance gap is primarily due to abstraction and planning errors rather than limitations of forward execution. We provide a failure-mode breakdown of backward planning in Appendix E. Deviation is the dominant failure mode, accounting for 60.4% of failed cases, followed by budget-exceeded cases at 31.7%, while looped plans account for only 7.9%. This further suggests that the main bottleneck lies in task-flow abstraction and predecessor prediction quality rather than pathological looping.

Forward search uses task-flow guidance together with replay to decompose long-horizon solution recovery into local one-flow searches; removing either component degen-

erates to random exploration. Experimental results show that this decomposition effectively improves the feasibility of long-horizon solution recovery. Successful trajectories typically span around 500 environment steps, confirming that the recovered solutions involve genuinely long-horizon behaviors rather than short-horizon coincidences. Appendix G provides representative successful trajectories illustrating how replay and task-flow alignment progressively reduce exploration difficulty.

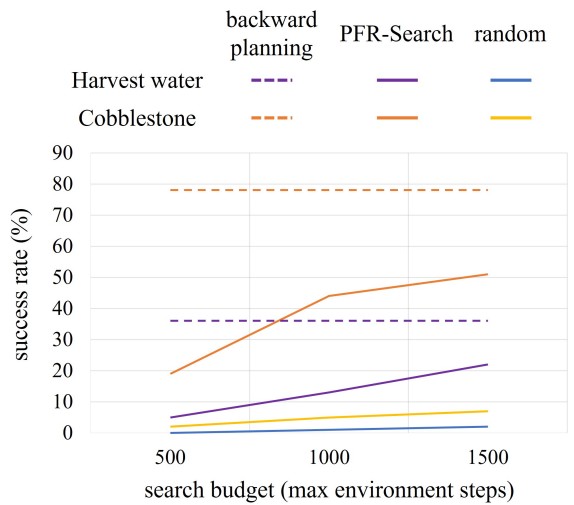

*Figure 5.* Success rate versus exploration budget for unguided random exploration and end-to-end PFR-Search.

**Effect of search budget.** Fig. 5 shows success rate as a function of the maximum exploration budget for two representative tasks, comparing unguided random exploration with end-to-end PFR-Search. Unguided random exploration increases only marginally with budget and remains at very low success rates, indicating that simply increasing interaction steps is insufficient for long-horizon tasks. In contrast, PFR-Search improves steadily as the search budget increases, demonstrating that task-flow guidance allows additional interaction to be effectively converted into task progress.

The dashed lines denote backward planning success rates.

*Table 4.* Effect of in-flow and cross-flow constraints. Numbers are changes in success rate (percentage points) relative to the full setting.

| Task | Backward Planning (BP) | | | End-to-End Search (E2E) | | |
|---|---|---|---|---|---|---|
| | w/o in-flow | w/o cross-flow | w/o both | w/o in-flow | w/o cross-flow | w/o both |
| Harvest water | 33.0 (-3.0) | 33.0 (-3.0) | 31.0 (-5.0) | 11.0 (-2.0) | 10.0 (-3.0) | 11.0 (-2.0) |
| Harvest log | 60.0 (-2.0) | 63.0 (+1.0) | 63.0 (+1.0) | 7.0 (-1.0) | 8.0 (+0.0) | 8.0 (+0.0) |
| Cobblestone | 76.0 (-2.0) | 73.0 (-5.0) | 69.0 (-9.0) | 43.0 (-1.0) | 40.0 (-4.0) | 39.0 (-5.0) |
| Iron ore | 48.0 (+1.0) | 43.0 (-4.0) | 47.0 (+0.0) | 7.0 (+1.0) | 6.0 (+0.0) | 7.0 (+1.0) |
| Harvest sand | 37.0 (+0.0) | 34.0 (-3.0) | 34.0 (-3.0) | 7.0 (-2.0) | 8.0 (-1.0) | 6.0 (-3.0) |
| Average | 50.8 (-1.2) | 49.2 (-2.8) | 48.8 (-3.2) | 15.0 (-1.0) | 14.4 (-1.6) | 14.2 (-1.8) |

As the search budget increases, the gap between end-to-end performance and backward planning narrows, suggesting that abstraction and backward planning accuracy play an increasingly important role in overall success. We further analyze the sensitivity of task-flow construction to the DBSCAN threshold $\epsilon$ in Appendix D.1.

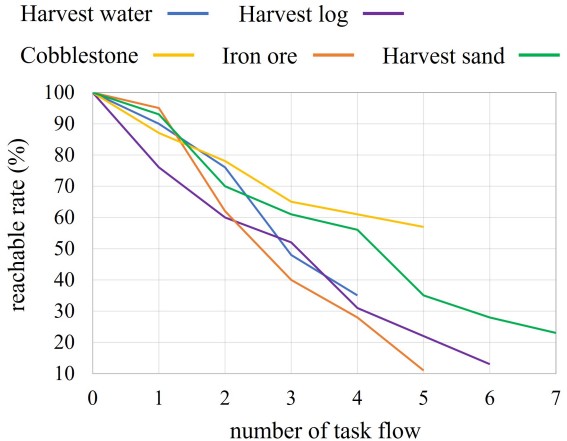

*Figure 6.* Flow-wise reachability along oracle task flows under forward search across five Minecraft tasks.

**Stage reachability.** Fig. 6 shows the reachability of each stage along an oracle task flow under forward search. Since the final stage corresponds to task completion, the curves characterize how success rates decay as more stages must be completed in sequence.

### 4.3. Effect of In-Flow and Cross-Flow Constraints

Tab. 4 analyzes the effect of removing in-flow and cross-flow constraints in backward planning (BP) and end-to-end search (E2E). In backward planning, removing either constraint degrades performance on average, with cross-flow having a larger impact (-2.8 pp) than in-flow (-1.2 pp). While cross-flow is generally helpful, we observe occasional variance where removing it slightly improves performance on specific tasks (e.g., Harvest log), suggesting task-dependent effects. Removing both causes the largest drop (-3.2 pp), indicating that the two constraints play com-

plementary roles in abstracting task-flow structure. End-to-end search shows smaller but consistent drops (-1.0 to -1.8 pp). This suggests that forward execution is partially buffered by replay and local search, but still benefits from the consistency imposed during flow construction.

Overall, these results show that in-flow and cross-flow constraints jointly induce a consistent and reliable flow abstraction, which directly supports backward planning and end-to-end search. Additional qualitative visualizations of the learned latent representations under different constraint settings are provided in Appendix F.

### 4.4. PFR-guided RL vs. Baseline RL

We compare PFR-guided RL with representative baselines commonly used in the video-driven Minecraft literature: (1) *VPT* (Baker et al., 2022), which uses large-scale video pretraining followed by behavioral cloning; (2) *STEVE-1* (Lifshitz et al., 2023), a multimodal model trained on paired video-text data; (3) *PTGM* (Yuan et al., 2024), which learns goal-reaching priors from large unlabeled gameplay; (4) *Director(PPO)* and *Director(MineCLIP)* (Fan et al., 2022), the latter providing dense vision-language progress signals. All methods are evaluated under the same downstream interaction budget of $10^6$ environment steps, using their standard reward formulations.

Across all five tasks, PFR-guided RL achieves competitive or superior performance by leveraging task-flow-aligned intrinsic rewards. This indicates that our method effectively extracts structured task knowledge from unannotated videos and converts it into a stable, actionable exploration signal.

## 5. Conclusion and Limitations

Many tasks do not require a reusable policy, and learning such policies typically relies on large amounts of expensive supervision. This motivates a shift from policy training to the direct recovery of solutions from unannotated task executions. We instantiated this perspective as PFR-Search, a policy-free framework that enables long-horizon solution recovery by decomposing the problem into recursive

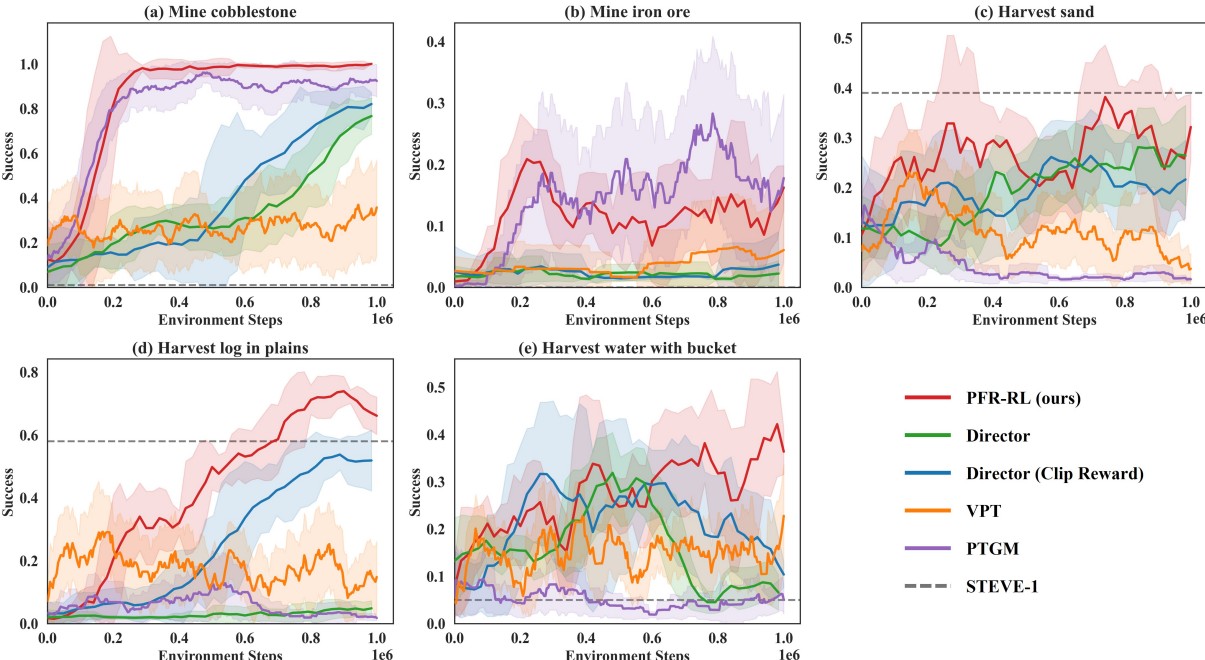

*Figure 7.* Performance comparison of PFR-guided RL with video-driven RL baselines on five Minecraft tasks.

backward planning and flow-guided forward search over a learned task-flow abstraction. We further show that the learned task-flow can be reused to provide intrinsic rewards that improve downstream policy learning.

Backward planning may fail on tasks with ambiguous state transitions, and forward search can be inefficient in environments with large branching factors or irreversible actions. PFR-Search also assumes that the video corpus contains sufficient and alignable task-progress structure. It does not require expert-optimal low-level actions, but unrelated, purposeless, or poorly covered videos may lead to unreliable task-flow abstractions. The current formulation is less suitable for tasks dominated by repeated or cyclic behavior, since the learned task-flow categories are designed to capture mostly directed progress. The reliance on discrete task stages further introduces a granularity tradeoff: coarse stages simplify cross-video alignment but make local reachability harder, whereas overly fine stages ease local search but can destabilize task-flow alignment across demonstrations. Finally, the current evaluation is limited to Minecraft; extending PFR-Search to larger or continuous action spaces would likely require stronger inter-stage transition mechanisms while preserving the high-level task-flow abstraction. Future work includes richer task-flow formulations, context-aware recurrent abstractions, and improved search strategies.

## Impact Statement

This paper offers a new perspective on decision making from unannotated data by shifting the focus from policy learning to direct solution recovery. Rather than training reusable policies, it explores how task structure in passive executions can be exploited to recover feasible solutions.

By introducing this viewpoint, the work may inspire alternative ways of using large unannotated video corpora for problem solving and long-horizon reasoning, potentially reducing reliance on dense supervision and extensive interaction.

## Acknowledgements

The work was supported in part by the National Key R&D Program of China under Grant No. 2024YFB4707600, the Fundamental and Interdisciplinary Disciplines Breakthrough Plan of the Ministry of Education of China under Grant No. JYB2025XDXM210; the National Natural Science Foundation of China under Grant Nos. 62125305, U23A20339, 62573339, 62503380, and 52435010; the Natural Science Foundation of Shaanxi Province under Grant No. 2025SYSSYSZD-083; the State Grid Corporation of China Headquarters Science and Technology Project under Grant No. 52060025005B-439-ZN; the Shaanxi Provincial Science Fund for Distinguished Young Scholars under Grant No. 2025JC-JCQN-077; and the CIE-Tencent Robotics XRhino-Bird Focused Research Program.

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

## A. Related Works

### A.1. Backward Planning & Search

A number of prior works exploit backward structure to mitigate the difficulty of long-horizon exploration and sparse reward propagation (Narvekar et al., 2020; Ivanovic et al., 2019; Gupta et al., 2019). These methods differ substantially in how backward information is used, and can be grouped into two main categories: backward model-based generation and bidirectional value propagation. Below we summarize each direction and highlight how PFR-Search fundamentally differs from both.

**Backward Models and Reverse Dynamics for RL**. Several methods learn backward or reverse dynamics models to generate informative trajectories. Model-Based Offline Planning performs trajectory-level planning over static datasets (Argenson & Dulac-Arnold, 2021), and subsequent work introduces trajectory pruning to improve scalability and robustness (Zhan et al., 2022). Recall Traces learn a backtracking model that predicts predecessor states and rolls it backward from high-value states to earlier parts of the state space (Goyal et al., 2019). Reverse model-based imagination similarly learns a reverse dynamics model and synthesizes trajectories backward from good states to augment offline RL datasets (Wang et al., 2021).

Bi-directional model rollouts extend this idea by generating pseudo-demonstrations via backward imagination for imitation, followed by forward RL fine-tuning (Pan & Lin, 2022). Related directions also explore latent and trajectory-level planning for long-horizon control and skill composition (Rosete-Beas et al., 2023; Janner et al., 2021; Chen et al., 2021). Planning has also been combined with self-imitation and policy optimization to support long-horizon learning (Gupta et al., 2019; Luo et al., 2021). These approaches depend on action-labeled trajectories or learned dynamics models. In contrast, PFR-Search requires neither actions nor dynamics, and performs backward planning purely over task-flow categories extracted from unannotated videos.

**Bidirectional Value Propagation and Backward-Informed Exploration**. A second direction incorporates backward information at the value, policy, or curriculum level. Forward-backward RL jointly propagates values in both temporal directions to improve credit assignment (Edwards et al., 2018), and has also been applied to domain-specific long-horizon problems such as Sokoban (Shoham & Elidan, 2021). Backward Curriculum RL expands the set of start states backward from the goal region to construct easier curricula for progressive learning (Ko, 2023), while curriculum learning more broadly has been studied as a general mechanism for improving exploration and training efficiency in RL (Narvekar et al., 2020). Related exploration methods address sparse rewards through novelty-driven exploration, intrinsic motivation, and goal-conditioned or autotelic behavior (Ladosz et al., 2022; Aubret et al., 2023; Colas et al., 2022). Other works improve exploration through reward uncertainty, theoretical guarantees for simple exploration strategies, or foundation-model-guided exploration (Liang et al., 2022; Dann et al., 2022; Ma et al., 2025b).

While these methods leverage backward structure, curricula, or exploration bonuses to improve policy learning, they still operate within policy- or value-centric RL pipelines and require environment interaction. Across these backward-reasoning and exploration methods, task progress is typically mediated by action-labeled trajectories, value functions, learned dynamics, rewards, or interactive policy optimization. PFR-Search instead performs backward reasoning over video-derived task-flow abstractions, converting unannotated videos into a discrete flow graph that directly guides backward planning and forward search before any reusable policy exists. The PFR-guided RL extension introduces only a simple flow-aligned shaping reward for downstream policy learning, rather than a learned reward model.

### A.2. Learning from Unannotated Videos for Decision Making

A broad range of recent work investigates how unannotated or action-free videos can support decision making and control (Baker et al., 2022; Seo et al., 2022; Pan et al., 2025; Li et al., 2023). Although these approaches share the common goal of exploiting large-scale human demonstrations without relying on action labels, they differ substantially in how video information is used. Existing methods can be roughly grouped into three categories: video-based policy pretraining, video-derived representations and reward signals for RL, and video-driven discovery of procedural or task structure. Below we review each direction and clarify how PFR-Search fundamentally differs from all of them.

**Unannotated Video Pretraining for Policy Learning**. A substantial line of research leverages unannotated or action-free videos to improve policy learning, exploration efficiency, or visual representation quality in RL. Several methods pre-train policy priors from large corpora of unannotated online videos. Representative examples include VPT, which learns behavioral priors from massive YouTube gameplay videos before inferring actions via inverse dynamics and fine-tuning with RL (Baker et al., 2022), and STEVE-1, which uses video-behavior logs to train generative models capable of producing complex behaviors (Lifshitz et al., 2023). Closely related efforts further study how unlabeled experience or passive data can be exploited to extract reusable skills, exploration priors, or behavior abstractions (Li et al., 2023; Hu et al., 2023; Wilcoxson et al., 2025; Kim et al., 2024). Although these approaches successfully extract useful priors from large-scale videos, they remain fundamentally tied to policy learning and require action grounding or policy optimization. In contrast, PFR-Search uses unannotated videos solely to extract task-flow structure and performs policy-free planning without requiring action labels, inverse dynamics, or imitation learning.

**Video-Based Representation, World Models, and Reward Learning**. Another line of work uses unannotated or passive videos to learn representations, world models, or reward signals that benefit downstream RL. Action-free video prediction pretraining has been used to initialize latent dynamics models for model-based RL (Seo et al., 2022), VIP learns value-implicit visual representations from unannotated human videos (Ma et al., 2023), and Video-Enhanced Offline RL incorporates Internet-scale videos into world-model training (Pan et al., 2025). Video prediction models have also been used directly as reward functions by evaluating predictive likelihoods (Escontrela et al., 2023), while State-to-Go Transformers leverage offline video demonstrations to construct intrinsic reward estimates for downstream policy learning (Zhou et al.,

2023). More recent approaches derive dense progress rewards from passive videos, such as TimeRewarder, which predicts frame-wise temporal distance as a shaping signal (Liu et al., 2026).

Beyond explicitly video-based methods, related work on pixel-based RL learns reusable visual representations through contrastive, temporal, or self-predictive objectives (Laskin et al., 2020; Stooke et al., 2021; Schwarzer et al., 2021). Complementary model-based approaches learn world models or latent-space planners that support downstream control through imagined rollouts, hierarchical latent goals, or visually faithful dynamics prediction (Hafner et al., 2022; 2024; Alonso et al., 2024). Overall, these methods use visual observations or videos to provide representation features, dynamics priors, or reward/progress signals, but they do not construct explicit task-flow abstractions or enable policy-free planning. PFR-Search differs by converting videos into a discrete flow graph that directly guides backward planning and forward search, without learning dynamics models or reward functions.

**Video-Based Task Structure and Procedure Learning**. A third related direction aims to discover procedural or step-wise structure from instructional or egocentric videos. Examples include hierarchical or state-grounded procedural representation learning (Zhao et al., 2026), self-supervised step discovery and localization (Dvornik et al., 2023), weakly supervised procedure correlation across videos (He et al., 2024), and online action segmentation for procedural tasks (Shen & Elhamifar, 2024). Related efforts also investigate progress estimation, expert progression modeling, and unsupervised structure discovery from videos (Bruce et al., 2023; Cheng et al., 2024; Duncan, 2011). These methods uncover temporal organization in human demonstrations but remain perception-oriented: they do not build directed task-flow graphs, do not operate without temporal or semantic supervision, and do not support search or decision making. In contrast, PFR-Search induces a discrete task-flow structure from unannotated videos and uses it directly for policy-free backward planning and forward exploration, bridging the gap between video-derived structure and actionable planning.

Across all three directions, prior work uses unannotated videos either to obtain policy priors, to learn perceptual or dynamical models for RL, or to extract procedural structure for recognition. None of these approaches provide a mechanism for policy-free planning or for guiding exploration before any policy or model exists. PFR-Search uniquely uses unannotated videos to derive a non-parametric task-flow graph and performs backward planning and forward flow search directly over this structure, enabling a fundamentally different pipeline for solving long-horizon tasks; PFR-guided RL then optionally builds on this structure by

adding a flow-aligned reward for standard policy learning.

## B. Baseline Selection & Fairness

**Baseline selection.** For the reinforcement learning setting, we build PFR-guided RL on PPO (Schulman et al., 2017), trained solely with the task-flow intrinsic reward. We compare PFR-guided RL with baselines spanning the two dominant paradigms for leveraging videos in Minecraft: (1) video-pretrained policies such as VPT (Baker et al., 2022), (Lifshitz et al., 2023), and PTGM (Yuan et al., 2024), which learn action-level or progression priors from large-scale unlabeled demonstrations; and (2) reward-shaped RL methods such as Director (MineCLIP) (Fan et al., 2022), which use learned progress models to provide dense feedback in sparse-reward environments. We additionally include Director (PPO) (Fan et al., 2022) as a non-video baseline. All baselines use their standard reward formulations, and all methods, including PFR-guided RL, are evaluated under the same downstream interaction budget of $10^6$ environment steps per task.

**Fairness.** To initialize task-flow abstraction, PFR uses a small warm-start set of 50 clips with coarse stage labels to initialize the VQVAE, clustering, and backward planner; all remaining learning uses only unannotated videos. This warm-start accounts for less than 5% of the full corpus (50 out of 1200 clips) and is substantially smaller than the amount of supervised data used in prior video-driven methods, which typically rely on large-scale annotated corpora or pretrained progression models. This warm-start is analogous to the pretrained components used in the above baselines (e.g., video-pretrained policies or progress models), and none of these pretraining stages are counted in the RL training step budget.

## C. Implementation Details

This appendix provides additional details on video preprocessing, task-flow learning, backward planner training, forward search for PFR-Search, and intrinsic rewards and PPO training for PFR-guided RL. All reinforcement learning agents in our evaluation are model-free and are trained or executed under the same $10^6$ environment step budget per task. Unless otherwise specified, PFR-guided RL results are averaged over 6 independent runs with different random seeds. Forward search and backward planning success rates are each estimated over 100 independent trials. The full configurations of the five downstream Minecraft tasks are provided in Tab. 5.

*Table 5.* This table details the configuration for the five downstream tasks evaluated in Minecraft. For each task, the specified Language Description is used to calculate the MineCLIP reward and evaluate the STEVE-1 baseline. The Initial Tools column indicates the items provided to the agent at the beginning of each episode, and Max Steps defines the maximum episode duration.

| Task | Language Description | Initial Tools | Max Steps |
|---|---|---|---|
| Harvest log in plains | "Cut a tree." | – | 1000 |
| Harvest water bucket in plains | "Find water, obtain water bucket." | bucket | 1000 |
| Harvest sand | "Obtain sand." | – | 1000 |
| Mine cobblestone | "Obtain cobblestone." | stone pickaxe | 1000 |
| Mine iron ore | "Obtain iron ore." | stone pickaxe | 2000 |

## C.1. Video Processing and VQVAE Training

We use 1200 clips from the OpenAI Contractor Dataset (Baker et al., 2022), sampled at 10 frames per second. Each clip lasts between 20 and 60 seconds. A subset of 50 clips is annotated with coarse task-stage labels and is used only to warm start the representation learning modules. All remaining learning uses unannotated videos.

The VQVAE (Van Den Oord et al., 2017; Razavi et al., 2019) encoder consists of a convolutional network followed by a gated recurrent unit operating on short temporal windows. Each input window contains 10 frames with stride 2, covering approximately 1 second of video. The convolutional layers extract spatial features which are passed into a recurrent encoder to capture short-range temporal structure. The latent embedding dimension is 128 and the GRU hidden dimension is 512. The codebook contains 128 entries. The model is trained in two phases: 2 epochs of unsupervised reconstruction pretraining followed by 10 epochs of warm-start training on the 50 annotated clips. No principal component analysis or other preprocessing is applied.

The overall training objective combines the in-flow consistency loss, cross-flow consistency loss, and the backward-prediction loss, weighted by coefficients $\alpha$ and $\beta$ respectively. Unless otherwise specified, we set $\alpha = 0.1$ and $\beta = 0.2$ in all experiments.

For clustering, we use DBSCAN to group latent vectors into discrete task-flow categories. DBSCAN is chosen because it does not require pre-specifying the number of clusters and can naturally handle noise and variable-density structure in long-horizon task videos. We apply DBSCAN directly to the learned latent vectors using cosine distance. The neighborhood radius $\epsilon$ is selected based on a small validation sweep, and we analyze sensitivity to $\epsilon$ in Appendix D.1.

DBSCAN is treated as a non-differentiable module. During training, clustering assignments are not backpropagated through: gradients from the in-flow and cross-flow objectives update only the encoder and backward planner, while

cluster assignments are held fixed. In practice, we periodically re-run DBSCAN on the updated latent representations to refresh cluster assignments, which are then used as fixed categories in the subsequent optimization phase.

Latent vectors from all video frames are flattened and aggregated across devices using distributed communication before clustering. No dimensionality reduction or normalization is applied beyond the encoder output. The resulting clusters define the discrete task-flow categories used for backward planning and forward search. Points classified as noise by DBSCAN are discarded and not used for task-flow construction.

## C.2. Clustering and optimization protocol.

DBSCAN provides discrete cluster assignments but is not differentiable. We therefore do not backpropagate gradients through the clustering procedure. Instead, after each clustering stage, we compute a prototype for each cluster as the mean of all latent vectors assigned to that cluster. We then define soft assignment probabilities $P(c_k \mid z)$ based on the cosine similarity between a latent vector $z$ and all cluster prototypes:

$$P(c_k \mid z) = \frac{\exp(\cos(z, \mu_k)/\tau)}{\sum_m \exp(\cos(z, \mu_m)/\tau)},$$

where $\mu_k$ denotes the prototype of cluster $c_k$ and $\tau$ is a temperature parameter. These soft assignments are used to evaluate the cross-flow consistency loss, and gradients are propagated only through the encoder parameters, while cluster assignments and prototypes are treated as fixed within each clustering stage.

Training alternates between updating the representation model under the reconstruction and consistency losses, and re-running DBSCAN on the updated latent space to refresh cluster assignments and prototypes. This alternating procedure allows the latent space to progressively align with the induced task-flow structure without requiring differentiable clustering.

## C.3. Backward Planner Training & Evaluation

**Training settings.** The backward planner is a 2-layer multilayer perceptron with hidden dimension 256 trained to predict the predecessor category for each flow category. Training data are constructed from the predicted flow labels of all videos.

We first obtain per-window label predictions using the trained VQVAE and clustering. These raw sequences are then cleaned by a debouncing procedure which requires at least 3 consecutive windows before accepting a state transition. Consecutive identical labels are merged into a single step. From each merged trajectory, we construct pairs of the form (current category, previous category). This produces a supervised dataset of adjacent category transitions.

The model is trained for 50 epochs using Adam with learning rate $10^{-3}$, batch size 512 and cross entropy loss. During inference, the model outputs a probability distribution over predecessor categories from which we sample after masking the current category.

**Evaluation protocol.** At test time, we first extract a set of task-flow sequences from held-out demonstrations and treat them as oracle task flows for the task. The flow category of the initial state and that of the final state are then provided to the backward planner, which infers a predecessor sequence to construct a candidate backward plan. We consider backward planning successful if the inferred category sequence matches any oracle task-flow sequence extracted from the demonstrations.

## C.4. Forward Search

Forward search evaluates whether the learned task-flow structure is sufficient for policy-free execution. The agent attempts to reach each planned category using a purely random policy. This design isolates the contribution of the flow structure by avoiding any learned control signal.

Forward search is organized as a sequence of one-flow searches. Each one-flow search is allocated a budget of 200 environment steps to reach the next target flow category. If the target is not reached within this budget, the agent rewinds to the current anchor state and begins a new attempt. When the number of replays from a single anchor exceeds 10, or when the total number of environment steps across all replays exceeds 1000, the one-flow search is terminated and considered a failure.

Search also terminates early if the environment reports task success or failure. Bad terminal states trigger rewinding rather than immediate termination. A forward search episode is considered successful if all target categories along the planned flow are reached within the step and replay limits.

## C.5. Intrinsic Rewards for PFR-guided RL

We define a task-flow-aligned intrinsic reward based on the extracted task flow. At each timestep, the agent receives +1 when its predicted flow category matches the next target category in the task flow, -1 when it enters a category outside the admissible set (neither the current category nor the next target category) and -0.1 as a per-step penalty to discourage unnecessary wandering. Completing the task yields an additional reward of +5.

During PFR-guided RL, PPO (Schulman et al., 2017) is trained using this intrinsic reward in place of the environment reward.

## C.6. PPO Training

The PPO agent receives $128 \times 128$ RGB observations and outputs actions in a 12-dimensional discrete action space. Training uses 4 parallel environments, learning rate $10^{-4}$, batch size 128, discount factor 0.999, GAE lambda 0.95, clipping range 0.2 and entropy bonus 0. Each task is trained for $10^6$ timesteps. These values follow the standard training configuration used in Minecraft environments.

## C.7. Baseline Training Details

All baselines are model-free reinforcement learning methods and are trained with the same $10^6$ environment step budget per task.

Video-pretrained policy baselines load official pretrained models: VPT follows (Baker et al., 2022), STEVE-1 follows (Lifshitz et al., 2023), and PTGM follows (Yuan et al., 2024). Reward-shaped RL baselines follow their original configurations: Director(PPO) and Director(MineCLIP) both follow (Fan et al., 2022), where Director(MineCLIP) uses the MineCLIP progress signal and Director(PPO) uses the environment reward. In all cases, we perform finetuning or training exactly as specified in the official implementations to ensure fair comparison with PFR-guided RL.

# D. Additional Ablations

## D.1. Sensitivity to DBSCAN Threshold $\epsilon$

The DBSCAN threshold $\epsilon$ controls the granularity of the learned task-flow categories. If $\epsilon$ is too small, semantically similar states may be split into fragmented categories; if it is too large, distinct task stages may be merged. We therefore evaluate whether PFR-Search is sensitive to reasonable changes in $\epsilon$ around the default value 0.2. Tab. 6 reports backward planning (BP) and end-to-end (E2E) success rates on two representative long-horizon tasks.

Across the tested range, the default value $\epsilon = 0.2$ gives the best or tied-best performance, while nearby values lead

*Table 6.* Sensitivity to DBSCAN threshold $\epsilon$. We sweep $\epsilon$ around the default value 0.2 while keeping all other settings fixed. All numbers are success rates (%) over 100 trials.

| $\epsilon$ | Harvest water | | Cobblestone | |
|---|---|---|---|---|
| | BP SR | E2E SR | BP SR | E2E SR |
| 0.1 | 30.0 | 8.0 | 66.0 | 32.0 |
| 0.2 | 36.0 | 13.0 | 78.0 | 44.0 |
| 0.3 | 34.0 | 13.0 | 75.0 | 42.0 |

only to moderate changes. This suggests that the learned task-flow abstraction is not overly sensitive to small variations in clustering granularity.

### D.2. Effect of Unlabeled Video Coverage

PFR-Search relies on unannotated task-execution videos to recover task-flow structure. A natural question is whether performance depends on the amount of available video coverage. To examine this, we reduce the number of unannotated clips used for task-flow learning from 1200 to 600 and 300, while keeping the evaluation protocol fixed. Tab. 7 reports the resulting backward planning success rates.

*Table 7.* Effect of unlabeled video coverage on backward planning success rate. Each result is evaluated over 100 backward-planning trials.

| Task | 1200 clips | 600 clips | 300 clips |
|---|---|---|---|
| Harvest water | 36.0 | 28.0 | 16.0 |
| Harvest log | 62.0 | 54.0 | 27.0 |
| Cobblestone | 78.0 | 71.0 | 37.0 |
| Iron ore | 47.0 | 33.0 | 20.0 |
| Harvest sand | 37.0 | 31.0 | 20.0 |
| Average | 52.0 | 43.4 | 24.0 |

Reducing the unlabeled video set from 1200 to 600 and 300 clips decreases the average backward-planning success rate from 52.0% to 43.4% and 24.0%, respectively. This confirms that PFR-Search depends on sufficient task-progress coverage, while also showing predictable degradation rather than abrupt collapse.

### E. Backward Planning Failure Modes

To better understand the remaining gap between learned-flow and oracle-flow performance, we analyze failed backward-planning attempts. We categorize failures into three types: budget-exceeded failures, where the planner does not reach the initial category within the planning budget; looped failures, where the planner repeatedly cycles through categories; and deviated failures, where the generated category sequence diverges from a valid task-flow path. Tab. 8 reports the resulting failure statistics.

Deviation is the dominant failure mode, accounting for 60.4% of failed cases, followed by budget-exceeded cases

*Table 8.* Backward planning failure-pattern statistics. Task rows report failure counts; the last row reports the overall proportion of each failure pattern among all failed cases.

| Task | Budget Exceeded | Looped | Deviated |
|---|---|---|---|
| Harvest water | 22 | 5 | 37 |
| Harvest log | 10 | 3 | 25 |
| Cobblestone | 8 | 0 | 14 |
| Iron ore | 18 | 5 | 30 |
| Harvest sand | 18 | 6 | 39 |
| Overall ratio (%) | 31.7 | 7.9 | 60.4 |

at 31.7%, while looped plans account for only 7.9%. This suggests that the main bottleneck lies in task-flow abstraction and predecessor prediction quality, rather than pathological looping.

### F. Visualization of Latent Representation under In-Flow and Cross-Flow Constraints

#### F.1. PCA Visualization under In-Flow Constraint

We train four variants of the model using different combinations of the proposed in-flow constraints and visualize their latent representations via 2D Principal Component Analysis (PCA) projections (Abdi & Williams, 2010) (Fig. 8). These projections are used only to illustrate qualitative trends in latent organization and are not sensitive to the choice of projection method.

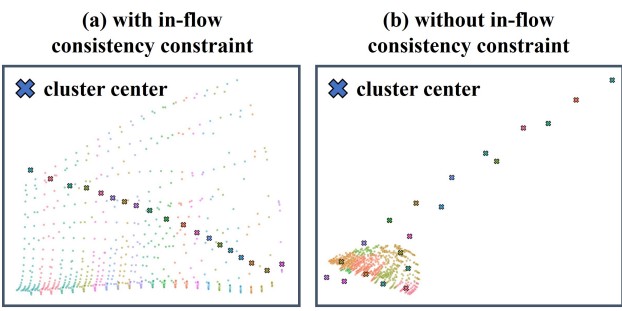

*Figure 8.* Visualization of in-flow constraint ablation. PCA projections of latent trajectories under different in-flow consistency settings.

With the directional constraint enabled, latent trajectories exhibit monotonic progression, forming clean streamline-like paths that reflect coherent task advance. Removing the directional constraint disrupts this structure, resulting in fragmented, irregular trajectories and scattered latent points. Across multiple runs, these qualitative patterns remain consistent and align with the observed stability of downstream stage prediction and backward planning.

## F.2. t-SNE Visualization under In-Flow Constraint

We provide a t-SNE visualization of the in-flow consistency constraint ablation study. As a non-linear dimensionality reduction technique, t-SNE focuses on the local structure of the high-dimensional latent space. The results are shown in Fig. 9, corroborating the findings from our PCA analysis and offering a complementary view of the manifold structure.

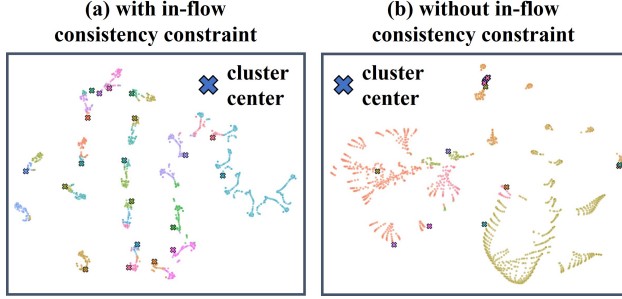

**(a) with in-flow consistency constraint**     **(b) without in-flow consistency constraint**

*Figure 9.* T-SNE visualization of the in-flow constraint ablation. Each plot corresponds to a different constraint combination, demonstrating how in-flow constraints shape the latent space.

The t-SNE visualizations further illustrate the effect of the in-flow consistency constraint on the learned latent space. (a) With in-flow consistency enabled, the embeddings organize into many well-separated local structures, where points form continuous, elongated patterns with a direction. This indicates a locally monotonic organization of the latent space, and cluster centers lie close to these structures, suggesting a decomposition into multiple coherent progression segments. (b) Without in-flow consistency, this organization breaks down. The embeddings are dominated by a few diffuse clusters with substantial internal mixing, where points spread in multiple directions and no clear progressive ordering is visible. The correspondence between cluster centers and surrounding embeddings also becomes less structured. Overall, the t-SNE plots suggest that in-flow consistency is critical for inducing a monotonic latent organization, in which local regions reflect coherent and directionally consistent state progression.

## F.3. t-SNE Visualization under Cross-Flow Constraint

To assess cross-demonstration consistency, we compare the full model (with both in-flow and cross-flow constraints) to an ablation variant without the cross-flow constraint. Fig. 10 visualizes the latent trajectories of two held-out demonstrations of the same task using t-SNE.

With the cross-flow constraint enabled, the two demonstrations trace out similar manifold shapes, indicating that corresponding task stages across videos are mapped to nearby regions in the latent space. Without the cross-flow constraint, the two trajectories remain internally coherent

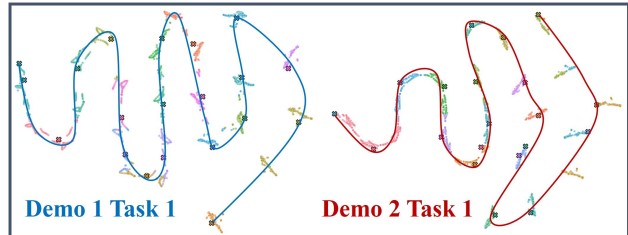

**(a) with corss-flow consistency constraint**

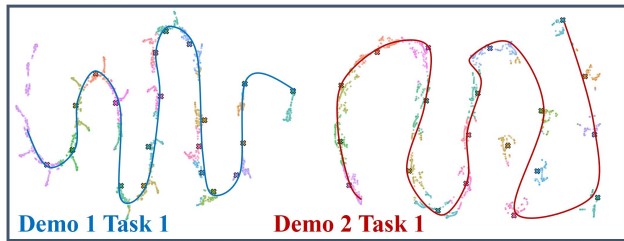

**(b) without cross-flow consistency constraint**

*Figure 10.* Visualization of cross-flow constraint ablation. t-SNE projections of two demonstrations of the same task, with and without cross-flow consistency.

due to in-flow constraints but diverge significantly from each other, reflecting inconsistent stage alignment across demonstrations. This qualitative difference is consistent with the downstream planning behavior observed in PFR-Search, where removing cross-flow consistency leads to unstable plans and more frequent rewinds.

## G. Forward Searching Demonstrations

We provide a visual demonstration of the Forward Searching algorithm's operational dynamics. Fig. 11 illustrates how our method guides an agent through two distinct, multi-step tasks: Harvest log in plains and Mine cobblestone. The horizontal axis represents frame samples generated during the one-flow search between adjacent categories, and the vertical axis denotes the planned task flow through backward planner, composed of sequential categories. The agent's objective is to follow this planned task flow, completing the search for each category in sequence. The figure illustrates exploration processes in two environments, from initial to goal frames. Notably, the first frame of each category is defined as an anchor state, if a search within a category fails, the agent returns to this anchor state through replay to restart the search, ensuring efficient exploration.

In subfigure (a) Harvest log in plains, the task flow consists of 6 categories. Category 1 (red) is the initial anchor state, where the agent starts near a tree. As the task proceeds, the agent undertakes frame-by-frame searches to transition through categories 2-5, each corresponding to distinct stages of interacting with the tree, e.g., moving closer, po-

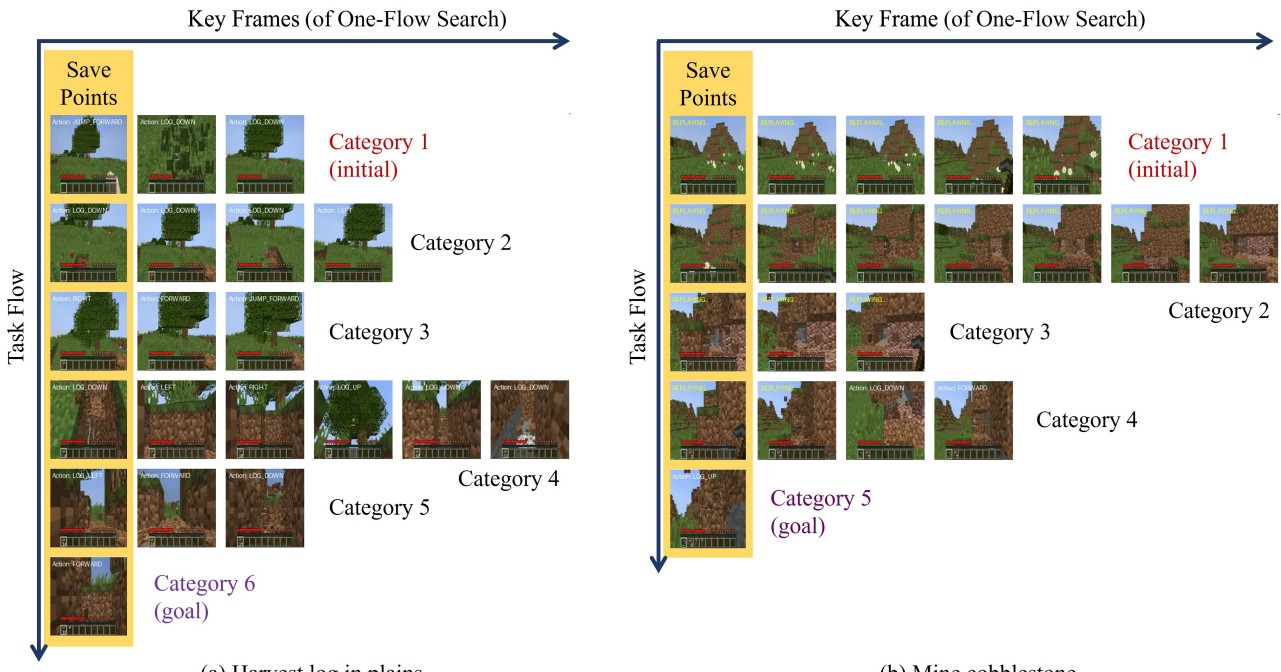

(a) Harvest log in plains                        (b) Mine cobblestone

*Figure 11.* Visualization of one-flow search in two Minecraft tasks. We show only representative key frames along one discovered task flow; each category aggregates many consecutive environment steps. In these examples, the illustrated flows contain 6 (Harvest log) and 5 (Mine cobblestone) categories, respectively. Anchor states (first frame of each category) enable replay-based exploration, with horizontal frame samples showing state transitions during inter-category search.

sitioning for logging. Category 6 (purple) is the goal category, marking the completion of log harvesting. Subfigure (b) Mine cobblestone follows a similar logic with 5 categories. Category 1 (red) is the initial anchor state, and Category 5 (purple) is the goal. The horizontal frame samples visualize the state transitions attempted during the search for each category, while the vertical anchor states anchor the start of each category's search.

These visualizations demonstrate the algorithm's capability to effectively model complex task flows and navigate an agent through them by leveraging the iterative one-flow search.

