# OpenReview forum: "Beyond Policy Training: Recursive Solution Search from Unannotated Videos"
_ICML.cc/2026/Conference — ICML 2026 regular_

### Official Review · Reviewer_NwF4 · 2026-03-08

**Soundness:** 3
**Presentation:** 3
**Significance:** 3
**Originality:** 3
**Overall Recommendation:** 5
**Confidence:** 3

**Summary:**

This paper proposes PFR-Search, a framework that recovers feasible solutions to long-horizon tasks directly from unannotated videos, without training any control policy. The core argument is simple: in many real-world settings you only need one working trajectory, so the expensive machinery of policy learning is unnecessary.

The method works in three stages. First, it learns a task-flow abstraction from unannotated videos by encoding frames with a VQVAE and clustering latents into discrete stages via DBSCAN, guided by two losses that enforce monotonic within-video progression and consistent stage alignment across demonstrations. Second, an autoregressive backward planner uses this abstraction to infer a high-level stage sequence from goal back to start. Third, forward search executes this plan using purely random actions, broken into local one-flow searches with an anchor-replay mechanism that preserves partial progress without any environment rollback. The same task-flow structure is also repurposed as intrinsic rewards for standard PPO training, which the authors call PFR-guided RL.
Experiments on five Minecraft tasks show end-to-end PFR-Search reaching 16% average success versus 1.8% for random exploration, with oracle flows pushing that to 27.8%. PFR-guided RL matches or outperforms several video-driven baselines under the same interaction budget.

**Compliance With Llm Reviewing Policy:**

Affirmed.

**Key Questions For Authors:**

N/A

**Limitations:**

yes

**Strengths And Weaknesses:**

### Strengths
The problem framing is novel. decoupling solution recovery from policy training is a meaningful conceptual contribution, not just a technical variation.

The system requires only 50 coarsely labeled warm-start clips, which is realistic.

The anchor-replay mechanism is an elegant solution to long-horizon exploration without privileged environment access. Ablations and visualizations are thorough and informative.

### Weaknesses
The absolute success rates are low, with some tasks barely above random, and the paper offers no clear path to improvement.

Backward planning succeeds only 52% of the time on average, directly capping end-to-end performance, yet failure modes are not concretely analyzed.

Relying on random actions for forward search is a fundamental limitation that will not scale to richer domains.

All experiments are in Minecraft, leaving generalization entirely open.

Computational costs are never discussed, and the claim that PFR-guided RL is competitive with VPT and STEVE-1 is difficult to interpret without controlling for total data budget.

---

> ### Author Rebuttal · Authors · 2026-03-31
>
> **Weakness 1: low absolute success rates**
>
> We agree that the current absolute success rates are still limited. For the central question of this paper, a key comparison is the gain over unguided random exploration, since the goal is to recover one feasible solution from unannotated task executions rather than to train a high-success reusable policy. At the same time, the oracle-flow comparison helps localize the bottleneck within the same recursive framework: the gap between learned flow and oracle flow indicates that the main performance loss comes from task-flow abstraction and backward planning quality, rather than from the recursive solution-recovery procedure itself. In this sense, random exploration establishes usefulness, while oracle flow clarifies where improvement is needed.
>
> **Weakness 2: backward planning failures are not concretely analyzed**
>
> We agree that this should be made more concrete. The paper already suggests that backward planning is more reliable on tasks with stable visual structure and predictable stage transitions, and less reliable when intermediate states are more ambiguous. To further clarify this, we analyzed failed backward-planning attempts by failure type, as shown in the table below. Deviation is the dominant failure pattern (58.0%), followed by budget-exceeded cases (34.4%), while looping accounts for a smaller portion (7.6%). This provides a clearer picture of how backward planning fails in practice and helps localize the main difficulty to task-flow abstraction and predecessor prediction quality.
>
> Table. Backward planning failure-pattern statistics. Task rows report failure counts; the last row reports the overall proportion of each failure pattern among all failed cases.
> | Task | Budget Exceeded | Looped | Deviated |
> |---|---:|---:|---:|
> | Harvest water | 22 | 5 | 37 |
> | Harvest log | 10 | 3 | 25 |
> | Cobblestone | 8 | 0 | 14 |
> | Iron ore | 18 | 5 | 30 |
> | Harvest sand | 18 | 6 | 39 |
> | Overall ratio (%) | 34.4 | 7.6 | 58.0 |
>
> **Weakness 3: random forward search scale to richer domains**
>
> We agree that this is an important question. Extending stage-based decomposition to more complex tasks involves a fundamental tradeoff: if stages are too coarse, reaching the next stage becomes difficult, while if they are too fine-grained, the induced task-flow becomes less stable across demonstrations. This is a basic challenge of the decomposition itself rather than a limitation unique to our method.
>
>
> **Weakness 4: generalization beyond Minecraft remains open**
>
> We agree that evaluation on a single benchmark leaves broader generalization open. We use Minecraft because it provides a practical common testbed for comparing several otherwise heterogeneous video-based methods under their intended settings: some rely on large-scale Minecraft video pretraining (VPT, PTGM), some use paired video-text supervision (STEVE-1), and some depend on MineCLIP-based progress shaping (Director with MineCLIP). Extending the full comparison to other environments would therefore require substantial re-annotation, retraining, or redesign of task-specific supervision. In this sense, the current paper should be viewed as an initial validation of the framework rather than a claim of broad cross-domain generalization.
>
> **Weakness 5: computational cost and total data budget fairness**
>
> We agree this should be clarified more explicitly. Our comparison is under the same **downstream interaction budget** of \(10^6\) environment steps, not matched **total pretraining cost** or **total data budget**. In fact, several baselines rely on substantially larger or more specialized pretraining resources, including large-scale Minecraft video pretraining (VPT, PTGM), additional labeled or mined text-video supervision (STEVE-1), and a MineCLIP reward model pretrained on MineDojo video data (Director with MineCLIP). By contrast, our method uses only 50 coarse warm-start clips and 1200 unlabeled clips. We will revise the paper to make clear that our claim is competitiveness under matched downstream interaction budget, not matched overall data or compute budget.

---

> > ### Author Rebuttal · Reviewer_NwF4 · 2026-04-04
> >
> > I do not have follow up questions.

---

### Official Review · Reviewer_pLLJ · 2026-03-11

**Soundness:** 3
**Presentation:** 3
**Significance:** 3
**Originality:** 3
**Overall Recommendation:** 4
**Confidence:** 3

**Summary:**

This paper proposed an intriguing approach to use unlabeled videos to solve tasks without learning a policy. The key idea is to extract a task-flow abstraction from unlabeled demonstrations by clustering latent representations of video frames and enforcing in-flow and cross-flow consistency. Based on this abstraction, the method performs a recursive backward–forward search: a backward planner first infers a sequence of intermediate task stages from the goal to the start, and a forward search then attempts to realize these stages through environment interaction. The framework can also be integrated with reinforcement learning by using the task-flow structure to define intrinsic rewards that guide exploration. Experiments on several Minecraft tasks show that the approach can recover feasible solutions from unlabeled videos and improve sample efficiency when used to guide RL.

**Compliance With Llm Reviewing Policy:**

Affirmed.

**Key Questions For Authors:**

1. **Dataset coverage.**
The framework appears to rely on sufficient coverage of task behaviors in the unlabeled video dataset. Have the authors studied how performance scales with the amount of video data? For example, what is the minimum dataset size required for the task-flow abstraction and backward planner to work reliably?

2. **Stage granularity.**
The effectiveness of the framework seems to depend on the granularity of the learned task stages. Typically, how long does each stage last (e.g., in terms of environment steps)?

3. **Scalability beyond discrete action spaces.**
The current forward search relies on random actions to reach the next stage. While this may be feasible in Minecraft due to the relatively small discrete action space, it may become difficult in domains with larger or continuous action spaces. Do the authors expect the approach to scale to such settings?

**Limitations:**

yes

**Strengths And Weaknesses:**

## Strength

1. The paper is well motivated, and the overall idea is interesting and novel.

2. Beyond directly enabling action planning, the proposed framework can also be used to provide intrinsic rewards for reinforcement learning policies, which may significantly improve exploration and learning efficiency.

3. Experimental results demonstrate superior performance compared to several baselines on five Minecraft tasks.

4. The paper includes additional analyses and visualizations that help improve understanding of the proposed method and its behavior.

## Weakness

1. **Clarity of method description.**
When the paper introduces cross-flow constraints in the method section, it mentions the BP (backward planner), which is only explained in later sections. This makes the section somewhat hard to follow. I think the clarity of this part could be improved by introducing BP earlier or briefly explaining it before referencing it.

2. **Relation to Koopman-style latent dynamics.**
The in-flow consistency idea is not entirely novel. At least, it seems related to the Koopman operator used in control. To my knowledge, Koopman-style formulations often regularize something like
$z_{t+1} = Kz_t$, where $K$ is a linear transformation that could be learned and is shared for the same trajectory,
while the in-flow consistency proposed here instead encourages local latent transitions to align with a global flow direction inferred from the trajectory. It may be helpful if the authors clarify this distinction and discuss how their formulation relates to Koopman-style latent dynamics.

3. **Dataset coverage assumptions.**
The method seems to implicitly rely on a dataset $D$ that covers a large portion of the behaviors required to solve the task. Since the task-flow abstraction and backward planner are learned from unlabeled videos, insufficient coverage could lead to missing or incorrect flow transitions. The current formulation of forward search also appears somewhat strict and may not tolerate errors or distributional shifts well. It would therefore be interesting to see experiments analyzing how performance scales with the amount of unlabeled video data, or what minimum dataset size is required for the method to work reliably.

4. **Sensitivity to imperfect task-flow plans.**
Related to the previous point, the proposed approach may also be sensitive to errors in the inferred task-flow plan. The paper argues that forward planning can provide meaningful intrinsic rewards for RL algorithms and therefore enable exploration. However, the current formulation of forward search enforces that the next transition must match either the current cluster or the planned next cluster. If the inferred plan is imperfect (for example due to limited video coverage), this constraint could potentially misguide exploration.

5. **Heuristic nature of forward search.**
The current forward-search procedure also appears somewhat heuristic and may not be the most general solution for harder tasks. For instance, it may struggle with tasks that require repetitive behavior, since such tasks may require revisiting the same cluster multiple times, which may not be possible if that cluster is also the terminal stage.

6. **Stage granularity tradeoff and use of video information.**
Finally, I feel the framework may not yet fully harness the information available in unlabeled videos. The method mainly learns high-level stage transitions from video data, but does not explicitly model how to transition between stages. As a result, forward search relies on random actions to reach the next stage. While this may be reasonable in the Minecraft setting due to the relatively small discrete action space, in other domains such as continuous control even reaching the next stage within a small number of random steps, say 10, may already be extremely unlikely.
In principle this may still work if the learned stages are sufficiently fine-grained (for example if each stage corresponds to only a few environment steps), since random exploration could still reach the next stage with reasonable probability. However, making the stage decomposition too fine-grained may introduce another challenge: the task-flow abstraction must align trajectories from different demonstrations, and very fine-grained stages may lead to highly inconsistent transition flows across demonstrations. This suggests there may be a tradeoff between stage granularity and search feasibility: if stages are too coarse, random exploration may struggle to reach the next stage, while overly fine-grained stages may make the transition flows across demonstrations highly inconsistent.

---

> ### Author Rebuttal · Authors · 2026-03-31
>
> **Weakness 1. Clarity of method description**
>
> We agree and will improve the presentation for clarity. We will introduce the backward planner earlier, or briefly explain it before it first appears in the cross-flow description.
>
> **Weakness 2. Relation to Koopman-style latent dynamics**
>
> We thank the reviewer for pointing out this connection. Our in-flow consistency is only broadly related to Koopman-style latent dynamics, since both impose temporal structure in latent space. The key difference is that Koopman-style methods learn a shared transition operator, whereas our in-flow term does not model latent dynamics at all. Instead, it is a trajectory-specific geometric regularizer that aligns local latent transitions with the global direction \(z_T - z_1\) of the same execution. We will clarify this distinction more explicitly in the revision.
>
> **Weakness 3. Dataset coverage assumptions**
>
> We agree that dataset coverage is an important assumption of the current formulation, since missing behaviors or transitions can directly affect both task-flow abstraction and backward planning quality. To examine this, we ran an additional data-scaling ablation by reducing the unlabeled video set from 1200 to 600 and 300 clips and measuring backward planning success. The average success rate drops from 50.82\% to 43.32\% and 23.92\%, respectively, showing that the method is indeed sensitive to coverage, but degrades in a clear and predictable manner rather than collapsing unpredictably. We will include this result and clarify the coverage assumption more explicitly.
>
> | Task | 1200 videos | 600 videos | 300 videos |
> |---|---:|---:|---:|
> | Harvest water | 34.5 | 28.2 | 15.8 |
> | Harvest log | 62.1 | 53.6 | 27.1 |
> | Cobblestone | 77.9 | 70.5 | 37.0 |
> | Iron ore | 46.5 | 33.0 | 20.2 |
> | Harvest sand | 33.1 | 31.3 | 19.5 |
> | Average | 50.82 | 43.32 | 23.92 |
>
> **Weakness 4: sensitivity to imperfect task-flow plans**
>
> We agree that the method is sensitive to the quality of the inferred task-flow plan, and an imperfect plan can indeed misguide both forward search and task-flow-based RL signals. This is exactly why the paper includes oracle-flow comparisons: the gap between learned-flow and oracle-flow performance reflects the impact of abstraction and planning errors. In this sense, plan quality is an explicit assumption of the current formulation rather than an overlooked issue. We will clarify this dependency more explicitly in the revision.
>
> **Weakness 5. Heuristic forward search**
>
> The recursive prototype is not merely heuristic, but a structured solution-recovery procedure: once backward planning recovers an ordered task-flow decomposition, forward search only needs to connect adjacent stages through local reachability. Repeated-stage ambiguity is partly alleviated by in-flow consistency, which encourages the representation to reflect task progress and reduces the chance of mapping repeated states at different progress positions into the same category. We agree that such cases remain challenging and could be further improved by context-aware sequence models.
>
> **Weakness 6: stage granularity and search feasibility tradeoff**
>
> We agree that there is an important trade-off here. If stages are too coarse, forward search may struggle to reach the next stage; if they are too fine-grained, the induced task-flow can become less stable across demonstrations. The current paper studies this framework in a regime where this tradeoff remains manageable, and the simple forward executor helps reveal the role of the learned stage structure more directly. We agree that explicit modeling of inter-stage transitions, or adaptive control of stage granularity, would be important directions for extending the framework to more difficult domains such as continuous control.
>
> **Q1. Dataset coverage**
>
> Please refer to weakness 3
>
> **Q2. Stage granularity**
>
> In the main paper, Table 1 reports the average number of backward-planned stages and Table 2 reports the successful trajectory length. Based on these two values, the average stage length can be approximated as: Harvest water (\(\sim 124\) steps), Harvest log (\(\sim 73\) steps), Cobblestone (\(\sim 95\) steps), Iron ore (\(\sim 122\) steps), and Harvest sand (\(\sim 59\) steps). Overall, this suggests that a typical learned stage spans several dozen to around one hundred environment steps.
>
> **Q3. Scalability beyond Minecraft / larger action spaces**
>
> We agree that this is an important question. More generally, this reflects a fundamental tradeoff in stage-based decomposition rather than a Minecraft-specific artifact: if stages are too coarse, reaching the next stage becomes difficult; if they are too fine-grained, the task-flow abstraction becomes less stable across demonstrations. Extending the framework to larger or continuous action spaces is therefore possible, but would likely require a stronger mechanism for inter-stage transitions while preserving the same high-level task-flow abstraction.

---

> > ### Author Rebuttal · Reviewer_pLLJ · 2026-04-04
> >
> > I appreciate the authors’ responses and their additional analysis. Overall, I think the proposed approach is currently most feasible in MDP (fully observable) settings with sufficient offline data. One case it may struggle with is tasks that require repetitive behaviors, which are effectively partially observable with single image input. While this limitation is also mentioned by reviewer aoAC (W8), I do not consider this a fatal flaw, as many video understanding and video-based control frameworks also operate on single-image inputs, effectively making a similar assumption.
> >
> > Overall, this work remains an interesting alternative to direct policy training, with potential for meaningful impact. As the authors have also acknowledged several existing limitations, I will keep my current score.

---

> > > ### Author Response · Authors · 2026-04-08
> > >
> > > Thank you very much for the thoughtful acknowledgement and for revisiting the paper after the rebuttal.
> > >
> > > We appreciate your careful and balanced assessment of the work, and we are glad that our responses helped clarify the scope and assumptions of the current formulation. We agree that extending the framework to more challenging settings will be an important direction for future work.

---

### Official Review · Reviewer_aoAC · 2026-03-12

**Soundness:** 2
**Presentation:** 2
**Significance:** 2
**Originality:** 2
**Overall Recommendation:** 2
**Confidence:** 4

**Summary:**

The authors propose a way to recreate behaviors described in unannotated videos.  They do so by learning frame-level latents, and a backwards planning model that can reverse-label a sequence of intermediate latent categories that lead from the initial state to the goal state.  The authors investigate the validity of this latent space and generated backwards paths over latent categories; firstly through Policy Free Recursive Search which recovers solutions through intermediate category-to-category steps via random action selections, and secondly by using the backwards category paths as implicit rewards to train a policy to follow such paths by selecting actions to achieve states that are encoded into the correct ordered sequence of categories.

**Compliance With Llm Reviewing Policy:**

Affirmed.

**Final Justification:**

I appreciate the rebuttal, but at the same time do not believe the authors have directly addressed many of my core concerns with the work.

At heart, the authors are presenting a **planning approach**.  They propose a high-level planner model in discrete space (the "backward planner"), and a plan follower (the "forward searcher"), terming this overall setting "policy-free".  This paradigm is **not new** - it is essentially performed by visual planning techniques where there is a high-level planner model in continuous space (the "video model planner") and a plan follower (the "inverse dynamics model") where there is also no "policy".  In both approaches the behavior is determined at the high-level plan level - where the only difference is what we plan over: discrete or continuous representations (e.g. continuous-valued latents that decode into RGB frames). The low-level module's job is simply to realize the high-level plan - whether it is through random action proposals or an IDM (they are interchangeable, you can implement "random-action policy-free search" with visual planning as well). Therefore I believe it is important to justify why we prefer **discrete** planning as opposed to **continuous** planning, whether through theory or through experimental comparisons to visual planning approaches (as an example "policy-free" technique). Visual planning also performs "horizon decomposition" but in continuous space (decompose behavior fixed horizon of future RGB frames) - the question is if we should do "discrete horizon decomposition" or "continuous horizon decomposition".

From this perspective, I do not agree with other reviewers that the "policy-free" aspect/solution discovery without training a policy is novel; this is already done by visual planning in continuous space. The **novel part is the *choice* of a discrete planning space** - and this is what the authors should focus on defending. Comparison between discrete and established continuous "policy-free" approaches is important particularly because working with discrete planning incurs many hacky design decisions (e.g. loop handling, environmental resets and replays, heuristic nature of forward search as noted by Reviewer pLLJ, etc.).  These, to be honest, are secondary concerns to the main research question at hand (discrete vs continuous planning) and can be justified if head-to-head experimental comparisons favor the discrete setting.

I suggest the authors compare against continuous "policy-free" approaches such as visual planning to defend their choice of discrete "policy-free" planning.  I believe such a comparison would clearly contextualize the proposed approach against other "policy-free" techniques, strengthening the core claims and intellectual contributions of the work, and making it a stronger submission for a future conference.  I will thus maintain my current rating.

**Key Questions For Authors:**

How can the model handle videos that showcase loops or repetitive behaviors (e.g. jumping jack repetitions, or repeatedly stirring a bowl)?  Such videos may break the assumption the authors make in Line 151 of "unannotated videos as recordings of task progression, where each execution reflects a directed and largely irreversible evolution of task state", or otherwise described by the authors as "in-flow consistency".

Does this work make assumptions on expertness/optimality of the video demonstrations?  It assumes that the videos from the demonstration set are already convertible to reasonable plans to imitate; however, in practice, they may be filled with suboptimal actions and it may not actually be optimal to have the low-level agent try to follow such a suboptimal plan.  This may particuarly be the case when this approach is extended to natural videos - there are many suboptimal or purposeless videos in general datasets.  Plan refinement may need to come from improving the high-level planner with respect to environment feedback.

How can backwards planning avoid loops when predicting predecessor categories?  As the authors describe, the self-category is ignored when sampling, to avoid staying in one category perpetually; but what about alternating between a category (a->b->a->b->...) in perpetuity, causing an unbreakable loop?

**Limitations:**

No negative societal impact or limitations need to be highlighted for this work.

**Strengths And Weaknesses:**

The main strength of this work is its novel approach towards using unannotated videos.

The main weakness of this work is that the policy-free recursive solution search, which is even the title of the work, is not that well motivated in terms of its usefulness.  It feels like the backwards plan generation (e.g. a high level planner) is more interesting, where using it to train a policy to follow such a plan by treating it as an intrinsic reward provider is more useful and practical.

Soundness: The submission does seem technically sound, but there aren't really proofs or anything to really say much more about this aspect.  Further justification on the design decisions of the method, particularly what alternatives were considered, and why they are the best choices for modeling what the authors want to would be helpful (e.g. the cross-flow loss term).

Presentation: The presentation is reasonable, but additional details on what the categories are, and how we can ensure they are semantically reasonable, could be provided in the main text.

Significance: The paper somewhat seems to address an important problem: how to utilize unannotated videos for decision-making.  However, it spends significant effort into describing a policy-free verification approach that does not appear to be that useful or practical for downstream decision-making.  In the end, it appears that the plan generator (backwards planner) and a trained policy to follow the plan is most practical for decision-making.

Originality: The work does have some originality (such as its policy-free recursive solution search, to this reviewer's knowledge) - but it is not always clear that such originality (e.g. the policy-free recursive solution search) is useful or enlightening.  It is not that surprising that we can perform random action search (e.g. policy-free) to eventually reconstruct a semantically similar sequence of frames, especially when we scale compute.  Indeed, with random policy search the authors state we can already achieve non-negative results (1.8%).  Then, doing so intermediate-step-by-intermediate-step but still with random actions intuitively will have higher success rate due to the problem breakdown (26%).  It is not clear what is particularly insightful or meaningful about this finding, or what is useful about it compared to direct policy supervision from the backwards policy.


Additional Weaknesses:
- The method is a bit hacky, especially its recovery strategy during forward searching.  It is stated that "Replay is implemented by re-executing the previously discovered action sequences from the environment initial state.  It does not assume any environmental rollback or state-setting capability."  However, this lightly assumes that the reset state is suitable (e.g. falls in the same category as the original initial state).  Furthermore, it is not obvious or guaranteed that replaying the same action sequence, when the original state has changed due to this reset, will result in the same sequence of previously-achieved categories to still be achieved.

- There are many moving components, such as a clustering algorithm that we rely on to be semantically meaningful.

- This work at a high level feels very similar to Visual Planning [3,4] (included in additional related works below).  In visual planning, there is a high-level plan generator (in this case, similar to the backwards plan generator) as well as a low-level plan follower (here, either the random-action policy-free search, or the policy trained to follow the plan).  The main difference is that the visual planner generates explicit visual latents/frames; here we use some cluster classes, and the IDM is trained on some small set of labeled interaction data.  The IDM difference is negligible; we can swap out pretraining on a fixed demonstration set with implementing the IDM as a policy learnable purely from online experience as it tries to follow the plan in an online manner (as done in this work).  Then, as the hierarchical structure of decision-making is the same with visual planning, the main question to answer in this work is: **why should we treat unannotated videos as sequences of some strange cluster classes as opposed to sequences of frames directly?**  In structuring the high-level plan, why would we prefer the plan to be classes at all?  The authors should consider comparing against other high-level planners that plan in other modalities (such as explicit visual planner approaches, or latent-space planners, etc.) to really highlight the design decision of cluster class plans being proposed in this work.  For example, a video planner can be compared against where the policy-free recursive solution search low-level executor will select random actions such that the resulting rendered frames sufficiently match the subsequent consecutive frame in the generated plan; or using the high-level video plan to train a policy by providing implicit rewards (the policy will learn to select actions that are sufficiently similar to the subsequent consecutive frame in the generated plan; aka learning the IDM from online experience alone without requiring any prior demonstration data).  This could be even done by evaluating against VIPER [1] at a frame-by-frame level; generate a subsequent frame to achieve and then have the policy either select random actions or learn to select actions until the environment progresses to a frame that is sufficiently similar (e.g. evaluated by some visual quality metric such as L2 distance or latent dissimilarity like DINO) to it before generating the subsequent goal frame.  The authors should even consider cutting out the classes and directly have the backwards planner plan in $z$ latent space; then have the policy-free search just perform random actions until the subequent encoded frame is sufficiently close to the planned $z$ (or have the task-flow guided RL policy learn such actions).  But at the heart of it, the authors should compare against other high-level-planner works, particularly to make the case for their choice of high-level-planning modality.

Additional related works:
- Video Generative Models as Zero-Shot Policy Supervision: Unannotated videos are directly used for decision-making by distilling into a policy (no a priori in-domain data is required, but self-collected experience can be used - as done here in task-flow guided RL) [2].
- Visual Planning: Unannotated videos are used through a generally pretrained video generative model for text-conditioned plan generation [3, 4]; a small amount of in-domain data is still needed to train an IDM to ground into executable actions (but may not be a "substantial" amount as noted in Line 112).  Adapted Visual Planners also use a small set of in-domain data to further improve visual planning performance [5], and can be used for self-improvement [6].
- LAPA: Unannotated videos can be used to large-scale pretrain latent actions, which can then be grounded into low-level robotic actions for decision-making (also uses a small amount of in-domain data) [7].
- SAGE: Not for decision-making explicitly, but some similar ideas on how to decompose unannotated videos into representative latents that describe a monotonic sequence [8].  In this work, Viterbi decoding is used; this may be of interest to the authors in thinking about alternate ways to structure monotonic sequences rather than or beyond directional alignment.


[1] Escontrela et al., Video Prediction Models as Rewards for Reinforcement Learning, NeurIPS 2023.

[2] Luo et al., Text-Aware Diffusion for Policy Learning, NeurIPS 2024.

[3] Du et al., Learning Universal Policies via Text-Guided Video Generation, NeurIPS 2023.

[4] Du et al., Video Language Planning, ICLR 2024.

[5] Luo et al., Solving New Tasks by Adapting Internet Video Knowledge, ICLR 2025.

[6] Luo et al., Self-Improving Loops for Visual Robotic Planning, ICLR 2026.

[7] Ye et al., Latent Action Pretraining from Videos, ICLR 2025.

[8] Zang et al., SAGE: A Unified Framework for Generalizable Object State Recognition with State-Action Graph Embedding, NeurIPS 2025 (Oral).

---

> ### Author Rebuttal · Authors · 2026-03-31
>
> **Main Weakness: unclear practical motivation**
>
> We agree that using structured information extracted from videos to support policy learning is an important and highly practical direction, and many existing works already study this setting. Our paper is motivated by a different question: whether one feasible solution can be recovered directly from unannotated task executions, without first learning a reusable policy. Accordingly, the policy-free search component is the main problem studied here, while the RL part is only a secondary evaluation interface. This setting is important because in some scenarios the goal is simply to obtain one workable solution, and training an additional controller may introduce unnecessary cost and supervision burden.
>
> **Weakness 2: insufficient justification of design choices**
>
> The main design choices are motivated by the formulation rather than ad hoc heuristics. To support backward planning over video-derived task flows, the learned abstraction must capture both coherent progression within each execution and transferable predecessor structure across demonstrations, motivating the in-flow and cross-flow constraints, respectively. Table 3 supports this empirically: removing either constraint degrades performance, and removing both causes the largest drop. We will make this rationale more explicit in the revision.
>
> **Weakness 3: insufficient explanation of categories**
>
> We agree this can be made more explicit in the main text. The categories are intended as discrete task-flow states, i.e., learned progress-level abstractions for backward planning, rather than raw visual clusters or manually defined subgoals. They are induced by VQVAE + DBSCAN and further shaped by the in-flow, cross-flow, and backward-predecessor constraints. We will clarify this more directly in the revision.
>
> **Weakness 4: unclear significance**
>
> Please refer to our response to the main weakness above. The significance of this work comes from studying a different objective from standard policy learning, namely direct recovery of one feasible solution from unannotated task executions without first training a reusable policy.
>
> **Weakness 5: unclear insightfulness**
>
> The key insight is that long-horizon solution recovery can be reduced to a sequence of finite local reachability problems by first recovering an ordered task-flow plan from unannotated task executions. Once the stage sequence is identified, forward search only needs to connect adjacent stages, which together yields a full solution trajectory. Our contribution is to show that this recursive structure can be induced directly from unannotated task executions, without downstream policy learning.
>
> **Weakness 6: replay under reset & trajectory drift**
>
> Replay restarts from the same initial condition and re-executes the saved action sequence, so it does not require rollback or arbitrary state setting. Its goal is not exact state restoration, but returning to a state within the same task-flow category as the discovered anchor, which allows some trajectory drift while preserving progress at the abstraction level.
>
> **Weakness 7: moving components**
>
> We agree that the method contains multiple components, but they are not arbitrary add-ons. In-flow enforces coherent progression within each execution, cross-flow aligns shared progression structure across demonstrations, the backward planner models predecessor relations for recursive search, and the 50 warm-up samples stabilize initialization. We will clarify this component logic in the revision.
>
> **Weakness 8: why plan over cluster**
>
> We plan over cluster classes because they are a discrete abstraction of the continuous latent space. This turns high-level planning from a continuous latent matching problem into a finite category-selection problem, making both backward planning and forward search much simpler and more structured.
>
> **Q1: handling loops**
>
> The next category in backward planning is sampled from non-current categories according to the learned predecessor distribution (as shown in Fig.3). Therefore, if the demonstrations contain transitions that exit the loop and continue task progress, the planner can in principle  jump out of the loop by sampling those transitions.
>
> **Q2: assumption on demonstration quality**
>
> Our method does not require expert-optimal demonstrations, because it does not imitate low-level actions directly. It only requires the videos to contain usable task-progress structure from which a reasonable task flow can be extracted. In this sense, suboptimal local actions are less critical than in policy imitation, as long as the demonstrations still reflect meaningful progress toward the goal.
>
> **Q3. loops in backward planning**
>
> Backward planning can enter loop-like transitions, but it does not run indefinitely (see Q1). We do not introduce an explicit loop-avoidance mechanism; instead, if the target category is not reached within the budget, the attempt is treated as a planning failure.

---

> > ### Author Rebuttal · Reviewer_aoAC · 2026-04-04
> >
> > It does not feel like the authors have substantially addressed the heart of my questions, and I would like to request additional clarification from the authors.
> >
> > To clarify my original question on replay under reset, I understand that the goal is not exact state restoration.  Nevertheless, re-executing the saved action sequence may not necessarily reproduce the same sequence of categories.  Then, progress may not actually occur at the "abstraction/category" level.
> >
> > To clarify my original question on why we should plan over clusters, I understand that cluster classes are a discrete abstraction of continuous latent spaces.  But this does not defend why it is a superior representation to learn planning over.  It may make backward planning and forward search simpler and more structured, but how can we ensure it is still as expressive as continuous latent space modeling?  And why would we want to use this backward planning and foward search at all in comparison to continuous space planning approaches (such as visual planning, for example, where an IDM may or may not be pretrained).  It is still not clear why we should "treat unannotated videos as sequences of some strange cluster classes as opposed to sequences of frames directly"?  Simply stating that discretization makes it easier for your proposed forward-backward approach is a chicken-and-egg/circular argument.  The authors have not defended why this approach is superior to visual planning or continuous space planning based approaches.
> > - To reiterate, there are direct similarities to visual planning: "In visual planning, there is a high-level plan generator (in this case, similar to the backwards plan generator) as well as a low-level plan follower (here, either the random-action policy-free search, or the policy trained to follow the plan). The main difference is that the visual planner generates explicit visual latents/frames; here we use some cluster classes, and the IDM is trained on some small set of labeled interaction data."  The authors should convincingly explain why we should use backward/forward with clusters instead of a potentially more expressive visual planner with an IDM.
> >
> > On handling loops - it is true that if a demonstration contains a transition to exit the loop than the planner can *in principle* jump out of the loop during sampling.  But the sampling procedure is random, and this planner has no real control over the amount of iterations the loop does at any given execution run.  It is also possible that the planner loops the same transitions a prohibitively high amount of times (being functionally "trapped" in the loop).  Would this not break the assumptions the authors pose in Line 151 that "each execution reflects a directed and largely irreversible evolution of task state"?  If videos depicted a loop that did not end, would also the unannotated data similarly break such a posed assumption of "in-flow consistency"?
> >
> > On demonstration quality - I understand that we do not require there to be expert-action-annotated demonstrations from which low-level actions are learned.  But when the authors state that "It only requires the videos to contain usable task-progress structure from which a reasonable task flow can be extracted." but is this not a requirement on demonstration quality?  Once again, in practice we may find that natural videos may follow suboptimal or unusable task-progress structure, as there are many suboptimal or purposeless videos in general datasets.  It still feels as if we require some level of demonstration quality in the training videos to make this proposed method work - and that is neither explored nor elaborated on (how to determine or generate a set of sufficiently-high-quality video demonstrations that enable the method to work).
> >
> > Overall, as a large majority of the questions posed have not been directly addressed (only misinterpreted), I cannot raise my score at this time.

---

> > > ### Author Response · Authors · 2026-04-05
> > >
> > > Thank you for the additional clarification. Because our previous rebuttal had to address many points under a strict space limit, some responses were necessarily brief. We therefore appreciate the opportunity to clarify these concerns more directly below.
> > >
> > > **On replay randomness**
> > > In our experiments, replay is deterministic at the implementation level: we fix the random seed and replay from the same initial condition, so the saved action sequence reproduces the same trajectory exactly. Therefore, the previously reached category sequence is also reproduced, so progress is preserved exactly at the abstraction level. We will make this condition explicit in the revision.
> > >
> > >
> > > **On discrete latent space**
> > > The recursive design comes from a simple observation: search difficulty grows sharply with task horizon. The core idea of recursive search is therefore to impose a decomposition on a long-horizon solution-search problem by introducing discrete intermediate stages. In our framework, the backward planner is used to infer this decomposition, i.e., a sequence of \(N\) stages connecting the start and the goal, and the forward search is then used to recover the solution one stage transition at a time. Assuming each stage transition has average horizon \(t\), the worst-case search-space size under the same action space \(|A|\) changes from \(O(|A|^{Nt})\) for end-to-end search to \(N \cdot O(|A|^t)\) under recursive decomposition.
> > >
> > > This also explains why we do not plan directly over raw frames or a continuous latent space. Planning directly over frames or visual trajectories is too fine-grained to provide the reduction we need, because it still leaves the planner effectively facing the full \(N\times t\)-step search problem at the raw-state level. Similarly, continuous latent-space planning does not by itself define such an intermediate decomposition, and typically requires an additional world model to predict or evaluate trajectories in continuous space. Visual trajectories are indeed more expressive, and such expressiveness is better suited to learning complex policies or grounding detailed execution. In the present policy-free setting, however, the main benefit comes from discrete horizon decomposition rather than representational expressiveness.
> > >
> > > **On handling loops**
> > > This issue is handled at both the mechanism and implementation levels.
> > >
> > > At the mechanism level, the in-flow constraint discourages loop-like trajectories by favoring directional progression over repeated oscillation. In practice, loops are relatively rare. Reviewer NwF4 raised a similar concern, and in our reply we analyzed backward-planning failure patterns:
> > >
> > > | Task | Budget Exceeded | Looped | Deviated |
> > > | --- | ---: | ---: | ---: |
> > > | Harvest water | 22 | 5 | 37 |
> > > | Harvest log | 10 | 3 | 25 |
> > > | Cobblestone | 8 | 0 | 14 |
> > > | Iron ore | 18 | 5 | 30 |
> > > | Harvest sand | 18 | 6 | 39 |
> > > | **Overall ratio (%)** | **34.4** | **7.6** | **58.0** |
> > >
> > > The table reports failure counts; the last row gives the proportion of each failure type. These results show that only 7.6\% of all failed cases are due to loops.
> > >
> > > At the implementation level, backward planning is budget-limited: if the planner does not reach the target stage within the allowed number of steps, the attempt is terminated and counted as a planning failure. More generally, if the raw data were dominated by loop-heavy temporal structure, our framework could handle such cases by explicitly modeling recurrent temporal patterns, for example with a Transformer or RNN.
> > >
> > > **On demonstration quality**
> > > The current formulation makes a data assumption, but not one of expert-action optimality. Our experiments use the OpenAI Contractor Dataset. In the original VPT data collection, contractors were initially asked to “play the survival mode of Minecraft like you normally would,” and more task-directed subsets were only added later. So this dataset should not be regarded as strictly expert-optimal demonstrations, and it is also widely used in Minecraft video/behavior learning, including VPT, STEVE-1, and PTGM.
> > >
> > > What our method requires is not optimal low-level actions, but sufficiently usable and alignable task-progress structure from which a stable task flow can be extracted. This is closely related to dataset-level progress coverage. To examine this, we reduced the unlabeled video set and measured backward planning success:
> > >
> > > | Task | 1200 videos | 600 videos | 300 videos |
> > > | --- | ---: | ---: | ---: |
> > > | Harvest water | 34.5 | 28.2 | 15.8 |
> > > | Harvest log | 62.1 | 53.6 | 27.1 |
> > > | Cobblestone | 77.9 | 70.5 | 37.0 |
> > > | Iron ore | 46.5 | 33.0 | 20.2 |
> > > | Harvest sand | 33.1 | 31.3 | 19.5 |
> > > | **Average** | **50.82** | **43.32**| **23.92** |
> > >
> > > The average success rate decreases from 50.82% to 43.32% and 23.92% as the amount of unlabeled video data is reduced.
> > >
> > > So our method does not require expert-optimal behavior, but it does assume a dataset in which task progress is sufficiently present and alignable across demonstrations.

---

### Official Review · Reviewer_8D7w · 2026-03-13

**Soundness:** 2
**Presentation:** 3
**Significance:** 3
**Originality:** 3
**Overall Recommendation:** 4
**Confidence:** 3

**Summary:**

This paper presents Policy-Free Recursive Search (PFR-Search), a framework to recover feasible solutions from unannotated videos without learning policies. The authors first use VQ-VAE and DBSCAN to extract task-flow categories from unannotated Minecraft videos.  They then learn a backward planner to auto-regressively predict a high-level roadmap from goal category. Given the roadmap, random actions are employed with anchor-based replay to complete the forward search. The authors further propose PFR-guided RL, which uses the task-flow structure to define intrinsic rewards for standard policy optimization. Experiments on five long-horizon Minecraft tasks show that PFR-Search achieves end-to-end success rate higher than random exploration but much lower than oracle flow, and PFR-guided RL is competitive with video-pretrained baselines under the same interaction budget.

**Compliance With Llm Reviewing Policy:**

Affirmed.

**Final Justification:**

The paper proposes a novel framing, recovering feasible solutions directly from unannotated videos without policy training. The authors implement it through a backward-forward recursive search framework. The methodology is coherent, the anchor-replay mechanism is clever, and the authors are honest about limitations.

The main weaknesses are the Minecraft-only evaluation, low success rates, and the inherent scalability limits of random forward search. These are real but represent boundaries of an initial validation rather than fundamental flaws.

The rebuttal addressed my concerns. The oracle-flow reward construction and baseline fairness are now clear. The warm-start ablation was informative. Framing the oracle-flow gap as an internal diagnostic that localizes the bottleneck to task-flow abstraction quality is convincing. Overall, the conceptual contribution is genuine and the work is technically sound within its stated scope. I maintain my updated score of 4.

**Key Questions For Authors:**

Please see some of my concerns in the weaknesses. Here are my additional questions:

1. What is the oracle-flow shaped reward in the PFR-guided reinforcement learning? Are rewards used in the baselines also shaped?
2. How important are the warm-start clips? What happens if we use more or fewer warm-start clips?

**Limitations:**

Yes

**Strengths And Weaknesses:**

**Strengths**

* The problem perspective is interesting and potentially valuable. While many prior studies center on learning representation and policies from video then adopt them for decision making, this work asks whether a video-centric approach can directly support search for feasible solutions.

* The methodology structure is clear and natural. The global-local decomposition into backward planning and forward search is intuitive and reasonable. The anchor-based replay mechanism is a clever way to preserve partial progress. The in-flow and cross-flow consistency loss are complementary and helpful in backward planning.

* The authors honestly present limitations. The 16% end-to-end success rate is reported without inflation, and the authors clearly acknowledge that backward planning is the bottleneck. It reflects that video-derived structure is useful, though not perfect (compared to random exploration).

**Weaknesses**

* The practical significance is limited by the narrow evaluation on only Minecraft environment, as well as the suboptimal performance, which remains far below the oracle-flow upper bound

* The in-flow loss encourages monotone movement along a clip-level direction, which raises the possibility that the representation is relying on a temporal shortcut instead of understanding semantic subgoals.

* The random forward search is simple but not efficient and very limited in large state and action spaces. The paper does not discuss this limitation quantitatively or propose any mitigation (e.g., learning a lightweight local policy prior).

* The intrinsic reward derived from video seems to help reinforcement learning. However, it is unclear how the oracle-flow shaped reward is constructed and whether other baselines receive comparably shaped rewards.

---

> ### Author Rebuttal · Authors · 2026-03-31
>
> **Weakness 1: narrow evaluation and far below oracle-flow**
>
> We acknowledge that evaluating on a single benchmark is a limitation. We use Minecraft because it provides a practical common testbed for comparing otherwise heterogeneous video-based methods under their intended settings: some rely on large-scale Minecraft video pretraining (VPT, PTGM), some use paired video-text supervision (STEVE-1), and some depend on MineCLIP-based progress shaping (Director with MineCLIP). Extending the full comparison to other environments would therefore require substantial re-annotation, retraining, or redesign of task-specific supervision, and could also make the comparison less fair, since several baselines are tightly coupled to Minecraft-specific pretraining resources or supervision setups.
>
> Regarding the gap to oracle flow, oracle flow is not an external competing method, but a control within the same framework: it uses the same recursive search procedure with oracle task-flow guidance. The gap to oracle flow therefore mainly indicates that the current bottleneck lies in learned task-flow abstraction / representation quality, rather than in the viability of the recursive solution-recovery framework itself. In other words, oracle flow localizes where performance is lost inside the framework, while the more relevant comparison for the paper's central question is the gain over unguided random exploration, since the goal is to recover one feasible solution from unannotated task executions.
>
> **Weakness 2: possible temporal shortcut from the in-flow loss**
>
> In-flow alone can indeed introduce this risk, which is exactly why we do not use it in isolation. A temporal shortcut can explain ordering within a single video, but it is much less able to explain a shared predecessor structure across different demonstrations that vary in length, speed, and visual details. Cross-flow consistency addresses this by requiring states from multiple videos to align to transferable predecessor relations between flow categories. This makes the full objective favor progression structure that is consistent across demonstrations, rather than features that only encode local temporal position within one execution. We will clarify this point more explicitly in the revision.
>
> **Weakness 3: inefficient search in large state/action spaces**
>
> Our formulation studies whether one feasible solution can be recovered without training an additional controller or policy prior, since in many settings the goal is not to learn a reusable policy but simply to obtain one workable solution. A stronger non-random controller with useful prior knowledge would typically require extra training, supervision, or modeling assumptions, which can become unnecessary overhead when the objective is only to recover a solution. Minecraft is already a challenging long-horizon domain, so the current results are not obtained in a trivial setting. We agree that stronger local mechanisms could further improve scalability, which is also part of the motivation for including PFR-guided RL.
>
> **Weakness 4 / Question 1: unclear oracle-flow reward and baseline fairness**
>
> The oracle-flow reward in PFR-guided RL is a task-flow-based shaped intrinsic reward that encourages reaching the next target flow category and penalizes deviation and unproductive steps. PFR-guided RL uses this reward in place of the environment reward, without anchor replay or rollback. Baselines follow their standard reward settings, e.g., Director (MineCLIP) uses the MineCLIP progress signal, while Director (PPO) uses the environment reward. Some baselines also rely on stronger supervision during pretraining: VPT uses labeled IDM training and pseudo-labeling on large-scale video data; STEVE-1 builds on VPT and MineCLIP with additional instruction-related supervision; and PTGM pretrains goal-based hierarchical models. By contrast, our method uses only 50 coarse warm-start clips and otherwise learns from unannotated task-execution videos. For downstream training, all methods are evaluated under the same \(10^6\) environment-step budget.
>
> **Question 2: unclear importance of warm-start clips**
>
> Warm-start clips are important for improving task-flow learning stability and backward planning performance. In an additional ablation with 0 warm-up clips, the average backward planning success rate drops from 52.0 to 24.8 across the five tasks. This shows that the warm-start set provides a substantial improvement in learning a more stable task-flow abstraction and backward planner. At the same time, performance remains clearly above zero without warm-start clips, indicating that useful task structure is still learned from the unannotated task-execution videos themselves.
>
> | Task | 50-sample warm-up | no warm-up |
> |---|---:|---:|
> | Harvest water | 36.0 | 24.0 |
> | Harvest log | 62.0 | 32.0 |
> | Cobblestone | 78.0 | 35.0 |
> | Iron ore | 47.0 | 21.0 |
> | Harvest sand | 37.0 | 12.0 |
> | Average | 52.0 | 24.8 |

---

> > ### Author Rebuttal · Reviewer_8D7w · 2026-04-04
> >
> > I appreciate the rebuttal and the additional clarifications. The authors have adequately addressed my main concerns for the current scope of the paper. The work still has clear limitations: most notably the evaluation is restricted to Minecraft, the random forward-search component may face scalability challenges in richer domains, and the current performance remains meaningfully below the oracle-flow upper bound. I view these as limitations of an initial validation rather than flaws that undermine the main contribution. I find the central direction of recovering one feasible solution directly from unannotated task-execution videos, rather than focusing only on policy learning, genuinely interesting and worthwhile. Overall, the rebuttal leaves me with a more positive assessment, and I am raising my overall score to 4.

---

> > > ### Author Response · Authors · 2026-04-08
> > >
> > > Thank you very much for the thoughtful acknowledgement and for taking the time to revisit the paper after the rebuttal.
> > >
> > > We are glad that our clarifications helped address your main concerns within the current scope of the work, and we also appreciate your balanced view of the current limitations. We agree that evaluation beyond Minecraft, scalability in richer domains, and closing the gap to the oracle-flow upper bound are all important directions for future work.

---

### Decision · Program_Chairs · 2026-04-30

**Decision:**

Accept (regular)

**Comment:**

This paper received ratings of 4, 2, 4, and 5, yielding an overall positive but mixed profile with one dissenting reject. After rebuttal, two positive reviewers and one strong accept reviewer indicated concerns were addressed and maintained their scores, while the reject reviewer remained unconvinced on novelty framing and comparison to continuous planning alternatives.

Reviewers agreed on key strengths: a novel and meaningful problem framing (solution recovery without policy training), a coherent recursive backward-forward framework, and useful empirical analyses added during rebuttal (coverage scaling, warm-start ablation, and failure-mode breakdown). The main remaining concerns are limited external scope (Minecraft-only), modest absolute success, and unresolved disagreement about whether the “policy-free” framing is sufficiently distinct from prior planning paradigms.

After reviewing the paper, rebuttal, and discussion, the AC agrees with the majority consensus and sees no sufficiently compelling reason to overturn it.

Final Recommendation: Accept